# MATEY: multiscale adaptive transformer models for spatiotemporal physical systems

## Abstract

Accurate representation of the multiscale features in spatiotemporal physical systems using vision transformer (ViT) architectures requires extremely long, computationally prohibitive token sequences. To address this issue, we propose two novel adaptive tokenization schemes that dynamically adjust patch sizes based on local features: one ensures convergent behavior to uniform patch refinement, while the other offers better computational efficiency. Moreover, we present a set of spatiotemporal attention schemes, where the temporal or axial spatial dimensions are decoupled, to evaluate their baseline computational and data efficiencies and to determine whether adaptive tokenization can improve this performance. We assess the performance of the proposed multiscale adaptive model, MATEY, in a sequence of experiments. Compared to a full spatiotemporal attention scheme or a scheme that decouples only the temporal dimension, we find that fully decoupled axial attention is less efficient and expressive, requiring more training time and model parameters to achieve the same accuracy. The experiments on the adaptive tokenization schemes show that, compared to a uniformly refined model, the proposed schemes achieve comparable or improved accuracy at a much lower cost. Finally, we demonstrat e in two fine-tuning tasks featuring different physics that models pretrained on PDEBench data outperform the ones trained from scratch, especially in the low data regime with frozen attention.

## 1 Introduction

Developing foundation models for physical systems is vital for energy generation, earth sciences, and power and propulsion systems. These models offer faster solutions than physics-based simulations and can generalize better across multiple systems than single-purpose AI approaches. However, their application to physical systems, often characterized by multiple sub-processes at different scales, is still in the early stages. For instance, fluid flowing around a cylinder creates a von Kármán vortex street, a highly dynamic flow with rapidly evolving vortices. Accurate solutions of such multiscale systems require a very high resolution representation to capture the most complex features across space and time. However, for scientific machine learning as for modeling and simulation, using very high resolutions to achieve accurate solutions incurs significant computational cost. This is particularly true for developing foundation models using vision transformer (ViT)-based architectures, as using the standard self-attention mechanism for extremely long spatiotemporal sequences can become prohibitively computationally expensive.

Efficient representation of multiscale features in high-resolution inputs has been an active research topic in computer vision. Three broad approaches can be characterized. First, multiscale models like Swin Transformer (Liu et al., 2021) and MViTv2 (Li et al., 2022) introduce multiple stages with decreasing resolution and increasing feature dimension for efficient hierarchical representations. Second, computational techniques have been developed that facilitate training on long sequences (e.g., sequence parallelism across GPUs (Jacobs et al., 2023)) or reduce the effective sequence length in the attention kernel (e.g., decomposing attention along axial directions (Ho et al., 2019)). Third, the actual sequence length can be directly shortened by pruning and merging tokens ((Haurum et al., 2023; Meng et al., 2022; Yin et al., 2022; Bolya & Hoffman, 2023)), though this strategy may lead to critical information loss (Liu et al., 2024).

These techniques have recently been adopted in scientific machine learning (SciML) for physical systems. For example, the atmosphere foundation model Aurora (Bodnar et al., 2024) uses Swin Transformer, while axial attention is applied by MPP (McCabe et al., 2023). Despite the progress, computational constraints remain a bottleneck, as existing approaches do not yet handle high-fidelity solutions of applications such as computational fluid dynamics, in which input sequences can easily exceed billions of tokens. More efficient algorithms are needed to enable the development of foundation models for multiscale multiphysics systems.

In this work, we develop a multiscale adaptive foundation model, MATEY (see Figure 1), that provides ~~two key algorithmic~~three contributions to address the challenges posed by spatiotemporal physical systems. First, we present a set of spatiotemporal attention schemes based on the axial attention (Ho et al., 2019) that differ in their decomposition of long spatiotemporal sequences and establish the cost in time-to-accuracy for decoupled spatiotemporal attention. Second, inspired by the adaptive mesh refinement (AMR) technique, we introduce adaptive tokenization methods that dynamically adjust patch sizes across the system based on local features, which provides as much as a $2\times$ reduction in compute for similar or higher accuracy, depending on the spatiotemporal attention scheme. Finally, we assess the fine-tuning performance of models pretrained on PDEBench (Takamoto et al., 2022) in two highly out-of-distribution settings, colliding thermals and magnetohydrodynamics (MHD), that include additional physical variables not included in pretraining and observe the pretrained models outperforming randomly initialized models.

## 2 Related work

**Scientific foundation models** Several research directions have been explored for building foundation models for physical systems, including MPP (McCabe et al., 2023) with PDEBench data, input augmentation with PDE system configurations (Hang et al., 2024), robust pretraining schemes (Hao et al., 2024), fine-tuning effectiveness investigations (Subramanian et al., 2024), and data-efficient multiscale ViT architectures (Herde et al., 2024). While these studies made remarkable progress, they do not directly address the issue of token sequence length, which becomes a computation bottleneck when applying ViTs to high dimension or high resolution data.

**Multiscale ViTs** While most multiscale ViTs achieve hierarchical representations via multi-stage attention blocks at different resolutions (e.g., MViTv2 (Li et al., 2022) and Swin Transformer (Liu et al., 2021)), there are a few focusing on tokenization schemes, such as (Yin et al., 2022; Fan et al., 2024; Zhang et al., 2024; Havtorn et al., 2023). A-ViT (Yin et al., 2022) improves efficiency for inference by removing unimportant tokens at inference; however, as these models still need to be trained on full set of tokens, it does not reduce training cost. The single-stage MSViT with dynamic mixed-scale tokenization (Havtorn et al., 2023) is the method most closely related to ours. MSViT employs a gating neural network (NN) to select which tokens to refine, together with an additional generalized batch-shaping loss (GBaS) term that constrains the computational cost (Bejnordi et al., 2020; Havtorn et al., 2023). However, in physical systems, we often have domain knowledge pinpointing areas of importance (e.g., interfaces of multiphase flows and flame fronts in combustion) with clear physical indicators, which the gating NN may not reliably capture. Moreover, its convergence speed and associated training cost are sensitive to the initialization of weights; e.g., the initial bias in MSViT was set to refine all tokens in (Havtorn et al., 2023). By contrast, the adaptive tokenization scheme in MATEY directly adjusts the patch sizes based on local feature scales, offering a simpler and more effective way to focus on areas of interest.

**Axial attentions** The quadratic scaling nature of attention makes it computationally prohibitive for extremely long token sequences from multidimensional systems. To address this challenge, (Ho et al., 2019) proposed the axial attention, which decomposes the full attention into a sequence of attention operations along each axis. It reduces the attention cost from $\mathcal{O}(N^{2d})$ to $\mathcal{O}(N^{d+1})$, for a given $d$-dimensional system with $N^d$ tokens. ViViT (Arnab et al., 2021) factorized the spatiotemporal attention into spatial- and temporal-dimensions for video classification. (McCabe et al., 2023) applied the axial attention in the Axial ViT (AViT) for spatiotemporal solutions of physical systems. While these spatiotemporal attention schemes can reduce the sequence length and hence the attention cost, their impact on accuracy in physical systems and on the performance of techniques like mixed-scale tokenization is unclear.

# 3   Methods

We propose multiscale adaptive foundation models, MATEY, to predict two-dimensional spatiotemporal solutions of multiple physical systems. The architecture of MATEY is illustrated in Figure 1. Given a sequence of $T$ past solutions of some physical system leading up to time $t$, MATEY predicts the solution at a future time $t + t_{\text{lead}}$ by learning from sequences of solutions for multiple physical systems. Specifically, MATEY learns a model $\mathbf{f_w}$ such that $\mathbf{u}_{t+t_{\text{lead}}} \approx \mathbf{f_w}(\mathbf{u}_{t-T+1}, \ldots, \mathbf{u}_t; t_{\text{lead}})$ by training parameters $\mathbf{w}$ to minimize the loss of the prediction from the solution sequence $\mathbf{U} = [\mathbf{u}_{t-T+1}, \ldots, \mathbf{u}_t]$ against the future solution with a lead time $\mathbf{u}_{t+t_{\text{lead}}}$. In the following paragraphs, we give detailed descriptions for each component in MATEY.

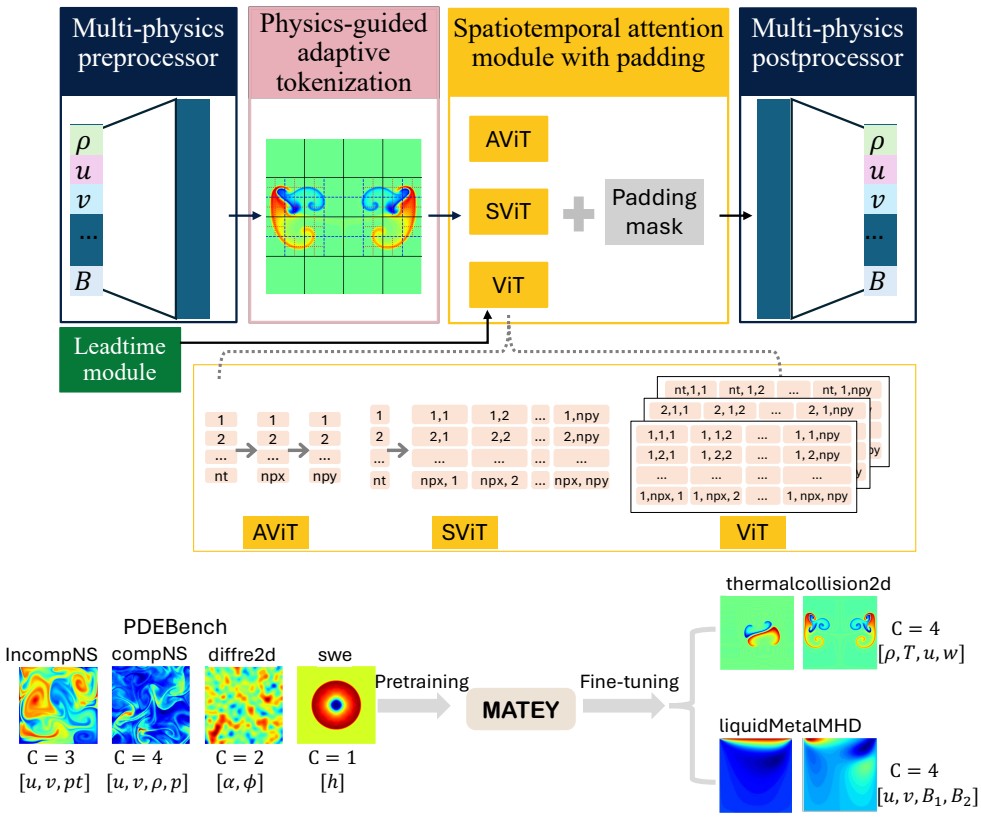

Figure 1: MATEY: multiscale adaptive foundation models for spatiotemporal physical systems.

## 3.1   Approach and architecture

**Multi-physics preprocessor, postprocessor, and training**   To accommodate multiple physical systems with different sets of variables at different spatial resolutions, we adopt the multi-physics preprocessor and postprocessor used in MPP (McCabe et al., 2023). For system $k$ with $C_k$ variables, the preprocessor first encodes solutions $\mathbf{u}_t(x, y) \in \mathbb{R}^{C_k}$ to a latent space $\mathbb{R}^{C_{\text{uni}}}$, where $C_{\text{uni}} \gg C_k$ is shared among all systems. Specifically, letting $H$ and $W$ denote the resolution in the $x$ and $y$ directions, respectively, the preprocessor encodes the solution $\mathbf{U}_k \in \mathbb{R}^{T \times H \times W \times C_k}$ of system $k$ into the unified latent representation $\mathbf{U} \in \mathbb{R}^{T \times H \times W \times C_{\text{uni}}}$. $\mathbf{U}$ is then tokenized into sequences $\mathbf{Z}^0 \in \mathbb{R}^{nt \times npx \times npy \times C_{\text{emb}}}$ in the tokenization module, which consists of convolutional blocks. Here $nt = T/p_t$, $npx = H/p_x$, and $npy = W/p_y$ are the number of patches in each dimension with prescribed patch size $[p_t, p_x, p_y]$. After passing through $L$ attention blocks, the input token sequence $\mathbf{Z}^0$ leads to the attention output $\mathbf{Z}^L \in \mathbb{R}^{nt \times npx \times npy \times C_{\text{emb}}}$. The last temporal snapshot of $\mathbf{Z}^L$ is then decoded in the postprocessor into the prediction $\mathbf{u}_{\text{pred}} \in \mathbb{R}^{H \times W \times C_k}$. In this work, the preprocessor is a linear map, the tokenization module is implemented as a convolutional block, and the final decoding postprocessor uses 2D transposed convolutional blocks. To train the model from solutions with different res-

olutions, we follow the approach in MPP by performing system-based sampling in the training process and fusing information from samples across different systems via multi-GPU training with PyTorch Distributed Data Parallelism (DDP) and gradient accumulation.

**Attention mechanisms — AViT, SViT, and ViT**   The standard ViT attention mechanism takes into account the attention across the entire set of spatiotemporal dimensions, which results in a high attention cost when extremely long spatiotemporal token sequences (e.g., from high-resolution spatiotemporal data) are considered. To address this issue, various factorized attention mechanisms have been proposed, such as AViT (Ho et al., 2019; McCabe et al., 2023) and a spatio-temporal decoupled attention (Arnab et al., 2021), referred to as SViT here. These attention mechanisms mainly consist of the same multihead self-attention (MHSA) and feed forward multi-layer perceptron (MLP) but differ in their attention block architecture. When $L$ attention blocks are cascaded, the standard attention block in ViT is given as

$$
\begin{aligned}
\widehat{\boldsymbol{Z}}^0 &= \boldsymbol{Z}^0 + \boldsymbol{E}_{\text{pos}}, \quad \boldsymbol{Z}^0 = [\boldsymbol{z}_1^0, \boldsymbol{z}_2^0, \dots, \boldsymbol{z}_N^0], \\
\boldsymbol{Z}^1 &= \text{MLP}(\widetilde{\boldsymbol{Z}}^1) + \widetilde{\boldsymbol{Z}}^1, \qquad \widetilde{\boldsymbol{Z}}^1 = \text{MHSA}(\widehat{\boldsymbol{Z}}^0) + \widehat{\boldsymbol{Z}}^0 + \text{MLP}(t_{\text{lead}}), \\
\boldsymbol{Z}^\ell &= \text{MLP}(\widetilde{\boldsymbol{Z}}^\ell) + \widetilde{\boldsymbol{Z}}^\ell, \qquad \widetilde{\boldsymbol{Z}}^\ell = \text{MHSA}(\boldsymbol{Z}^{\ell-1}) + \boldsymbol{Z}^{\ell-1}, \qquad \ell = 2, \dots, L
\end{aligned}
\tag{1}
$$

where $[\boldsymbol{z}_1^0, \dots, \boldsymbol{z}_N^0]$ denotes the full spatiotemporal token sequence of length $N$ with each token $\boldsymbol{z}_i^0 \in \mathbb{R}^{C_{\text{emb}}}$, $\boldsymbol{E}_{\text{pos}}$ is a positional embedding term, and each MHSA and MLP is followed by an `InstanceNorm1d` module. In ViT, the token sequence includes all spatiotemporal patches, meaning $N = nt \cdot npx \cdot npy$, resulting in an overwhelming cost of $\mathcal{O}((nt \cdot npx \cdot npy)^2)$ operations for attention. In contrast, SViT decouples the attention into $npx \cdot npy$ time-attention blocks and $nt$ space-attention blocks cascaded sequentially, as in "$\text{MHSA}_{\text{time}} \to \text{MHSA}_{\text{space}} \to \text{MLP}$",

$$
\begin{aligned}
\text{Time sequences:} \quad & \boldsymbol{Z}_i^{\ell-1} = \left[\boldsymbol{z}_{(i-1)\cdot nt+1}^{\ell-1}, \boldsymbol{z}_{(i-1)\cdot nt+2}^{\ell-1}, \dots, \boldsymbol{z}_{(i-1)\cdot nt+nt}^{\ell-1}\right], i = 1, \dots, npx \cdot npy \\
\text{Attention in time:} \quad & \boldsymbol{Z}_i^{\ell-\frac{1}{2}} = \text{MHSA}_{\text{time}}\left(\boldsymbol{Z}_i^{\ell-1}\right) + \boldsymbol{Z}_i^{\ell-1}, \quad i = 1, \dots, npx \cdot npy \\
\text{Space sequences:} \quad & \check{\boldsymbol{Z}}_t^{\ell-\frac{1}{2}} = \left[\boldsymbol{z}_t^{\ell-\frac{1}{2}}, \boldsymbol{z}_{t+nt}^{\ell-\frac{1}{2}}, \dots, \boldsymbol{z}_{t+nt\cdot(npx\cdot npy-1)}^{\ell-\frac{1}{2}}\right], \quad t = 1, \dots, nt, \\
\text{Attention in space:} \quad & \widetilde{\boldsymbol{Z}}_t^\ell = \text{MHSA}_{\text{space}}\left(\check{\boldsymbol{Z}}_t^{\ell-\frac{1}{2}}\right) + \check{\boldsymbol{Z}}_t^{\ell-\frac{1}{2}}, \quad t = 1, \dots, nt, \\
\text{Feed forward ML:} \quad & \boldsymbol{Z}^\ell = \text{MLP}\left(\widetilde{\boldsymbol{Z}}^\ell\right) + \widetilde{\boldsymbol{Z}}^\ell, \quad \ell = 1, \dots, L,
\end{aligned}
\tag{2}
$$

which reduces the MHSA cost to $npx \cdot npy \cdot \mathcal{O}(nt^2) + nt \cdot \mathcal{O}((npx \cdot npy)^2)$. The position embedding and the lead time MLP are omitted in (2) for simplicity. AViT further decomposes the space-attention in SViT into two axial directions following the same approach, which leads to a cost of $npx \cdot npy \cdot \mathcal{O}(nt^2) + nt \cdot npy \cdot \mathcal{O}(npx^2) + nt \cdot npx \cdot \mathcal{O}(npy^2)$. The decomposition in both AViT and SViT neglects some spatiotemporal correlations, and thus gives shorter token sequence length for each attention block, at the cost of introducing additional attention blocks. These extra attention blocks moderately increase the model size, as shown in Table 1. Note that within the same size category considered in Table 1, AViT and SViT are larger than ViT due to the additional MHSA, while AViT and SViT have similar sizes because AViT reuses the same attention blocks for different spatial directions. In MATEY, we implement the three attention mechanisms – AViT, SViT, and ViT – and evaluate their performance on test problems to study how the lost spatiotemporal correlations affect the quality of the solution and to assess the impact of decoupled attentions with additional attention blocks on the learning efficiency for multi-physics foundation models.

**Pretraining and fine-tuning**   We pretrain the models on PDEBench data, which includes five basic 2D systems: incompressible flows, compressible flows, turbulent flows, reaction-diffusion systems, and shallow water equations. We consider two fine-tuning cases: 1) colliding thermals between a cold and a warm bubbles from MiniWeather simulations (Norman, 2020) and 2) lid-driven cavity MHD flows (Fambri et al., 2023). We will release the two datasets upon paper publication. As discussed in detail in Appendix A.1, these fine-tuning datasets were selected to be meaningfully out-of-distribution, not only in flow regime but also in including thermal and electromagnetic components that are not represented at all in the pretraining data. Training was performed on the Frontier and Perlmutter supercomputers at the Oak Ridge Leadership Computing Facility (OLCF) and National Energy Research Scientific Computing Center (NERSC), respectively.

## 3.2 Adaptive tokenization methods

Smaller patch sizes are preferred for better representation accuracy, as ViTs can capture long-range correlations between patches well but lack inductive biases within patches. However, features in physical systems often cross multiple length scales and exhibit strong spatiotemporal inhomogeneities, such as mixing layers in ocean flows, interfaces in multiphase flows, and reaction fronts in reacting flows. Consequently, constant patch sizes that are small enough to provide good accuracy in the necessary regions of such systems result in impractically long token sequence lengths over the entire domain. To address this issue, we propose an adaptive ViT that dynamically adjusts the tokenization patch sizes according to local physical features. To maximize expressiveness, we start with coarse patching and identify the most complex patches in each sample based on a simple metric, such as the variance of local features. The identified patches are further refined to the sub-token-scale (STS) to improve representation accuracy. Adaptive patch size leads to patches of varying length across samples, which are handled with padding masks. Patch position and patch area bias are represented following the embedding method in (Bodnar et al., 2024). For simplicity, we describe the method using notations with two token-scale levels; however, the method generalizes naturally to any number of levels.

For a given solution field $\boldsymbol{u}_t \in \mathbb{R}^{H \times W \times C}$, tokenization at a constant patch size $[p_x, p_y]$ is achieved through a convolutional block and leads to a patch grid of size $(npx, npy) = (H/p_x, W/p_y)$. For adaptive tokenization, we apply varying patch sizes in space based on local complexity represented by the patch variance. For a solution $\boldsymbol{u}_t \in \mathbb{R}^{H \times W \times C}$ and an initial coarse patch size $[p_{x_1}, p_{y_1}]$, a variance tensor $\boldsymbol{v}_t \in \mathbb{R}^{npx_1 \times npy_1}$ ($npx_1 = H/p_{x_1}$ and $npy_1 = W/p_{y_1}$) is calculated from solutions inside each patch of the reshaped solution $\widetilde{\boldsymbol{u}}_t \in \mathbb{R}^{npx_1 \times npy_1 \times p_{x_1} \times p_{y_1} \times C}$ as

$$\boldsymbol{v}_t(i,j) = \frac{1}{C \cdot p_{x_1} \cdot p_{y_1}} \sum_{c=1}^{C} \sum_{k=1}^{p_{x_1}} \sum_{l=1}^{p_{y_1}} \left( \widetilde{\boldsymbol{u}}_t(i,j,k,l,c) - \frac{1}{p_{x_1} \cdot p_{y_1}} \sum_{k=1}^{p_{x_1}} \sum_{l=1}^{p_{y_1}} \widetilde{\boldsymbol{u}}_t(i,j,k,l,c) \right)^2, \tag{3}$$

where $(i,j)$ denotes the patch's coordinate on the $npx_1 \times npy_1$ grid. Patches with variance values greater than a prescribed threshold are then selected for further refinement at a smaller patch size. Specifically, let STS-IDs denote the index set of patches to be refined, then

$$\text{STS-IDs} := \{(i,j) | \boldsymbol{v}_t(i,j) > \gamma_{\text{sts}} \cdot \boldsymbol{v}_{t,\max}\}, \quad N_{\text{sts}} := |\text{STS-IDs}|, \tag{4}$$

where $\gamma_{\text{sts}} \in [0,1]$ is a user-specified hyperparameter, $\boldsymbol{v}_{t,\max}$ is the maximal variance among all patches, and $N_{\text{sts}}$ is the number of patches to be refined. The selected patches are refined to patches of a smaller size $[p_{x_{\text{sts}}}, p_{y_{\text{sts}}}]$, referred to as "STS tokens" in this work, where $\boldsymbol{Z}^0_{\text{sts},i} = \left[ \boldsymbol{z}^0_{\text{sts},1}, \boldsymbol{z}^0_{\text{sts},2}, \dots, \boldsymbol{z}^0_{\text{sts},p_{x_1}/p_{x_{\text{sts}}} \times p_{y_1}/p_{y_{\text{sts}}}} \right]_i$ ($i = 1, \dots, N_{\text{sts}}$). The STS tokens can be combined with the coarse tokens in two ways, as shown in Figure 2. In the first approach, referred to as "`Adap_Mul`" (for adaptive multi-resolution tokenization), we consider the coarse and STS tokens as separate sequences, passing through the attention blocks serially. In the second approach, referred to as "`Adap_Mix`" (for adaptive mixed-resolution tokenization), we replace the selected coarse patches with the sequence of STS tokens directly appended to the end of the sequence.

After spatiotemporal attention, the decoding of adaptive patch sequences into solution fields within the multiphysics postprocessor is performed using transposed convolutional blocks, tailored to each corresponding scale. For `Adap_Mul`, the patches at different resolutions/sizes are deconvoluted separately and then summed to the final output, $\widehat{\boldsymbol{u}}_t$. Specifically, for a coarse attention output $\boldsymbol{Z}^L_{\text{coarse}} = \left[ \boldsymbol{z}^L_1, \boldsymbol{z}^L_2, \dots, \boldsymbol{z}^L_{npx_1 \times npy_1} \right]$ and STS attention outputs $\boldsymbol{Z}^L_{\text{sts},i} = \left[ \boldsymbol{z}^L_{\text{sts},1}, \boldsymbol{z}^L_{\text{sts},2}, \dots, \boldsymbol{z}^L_{\text{sts},p_{x_1}/p_{x_{\text{sts}}} \times p_{y_1}/p_{y_{\text{sts}}}} \right]_i$ ($i = 1, \dots, N_{\text{sts}}$), "`Adap_Mul`" performs the following operations:

$$\begin{aligned}
\text{Reconstruction from coarse patches:} \quad & \widehat{\boldsymbol{u}}_t = \text{ConvTranspose2d}_1(\boldsymbol{Z}^L_{\text{coarse}}), \\
\text{Reconstruction from STS patches:} \quad & \widehat{\boldsymbol{u}}_{t,\text{sts},i} = \text{ConvTranspose2d}_2(\boldsymbol{Z}^L_{\text{sts},i}) \\
& \widehat{\boldsymbol{u}}_{t,\text{sts}} = [\widehat{\boldsymbol{u}}_{t,\text{sts},1}, \dots, \widehat{\boldsymbol{u}}_{t,\text{sts},N_{\text{sts}}}] \\
\text{Fusion of multi-resolution solutions:} \quad & \widehat{\boldsymbol{u}}_t[\text{STS-IDs}] = \widehat{\boldsymbol{u}}_t[\text{STS-IDs}] + \widehat{\boldsymbol{u}}_{t,\text{sts}}.
\end{aligned} \tag{5}$$

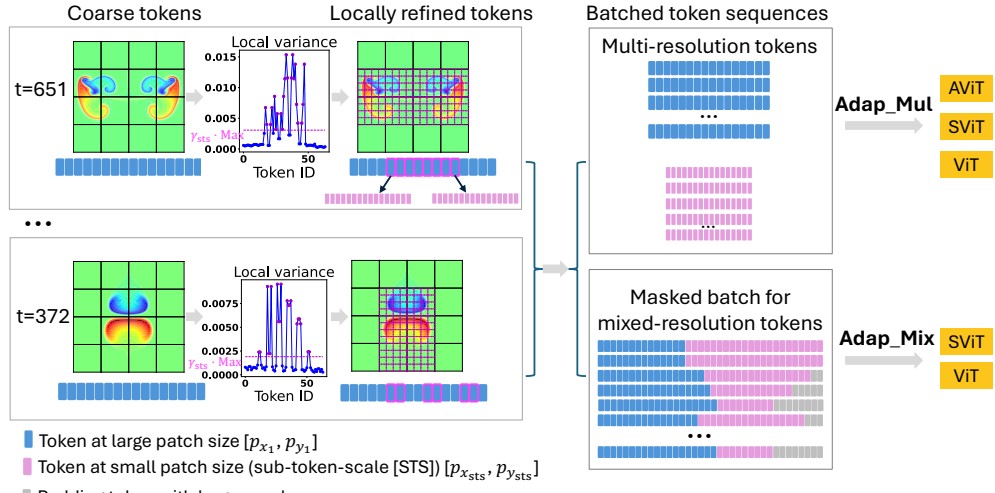

Figure 2: Adaptive tokenization that dynamically adjusts patch sizes based on local features. There are three essential parameters: $[p_{x_1}, p_{y_1}]$, $[p_{x_{\mathrm{sts}}}, p_{y_{\mathrm{sts}}}]$ and $\gamma_{\mathrm{sts}}$. The parameter $[p_{x_1}, p_{y_1}]$ denotes the initial coarse patch size, $[p_{x_{\mathrm{sts}}}, p_{y_{\mathrm{sts}}}]$ represents the refined patch size, and $\gamma_{\mathrm{sts}} \in [0, 1]$ determines which patches to refine. We select patches with local variances greater than $\gamma_{\mathrm{sts}}$ times the maximum variance across all patches (see Equation (4)).

On the other hand, the `Adap_Mix` approach fuses the coarse and STS patch sequences into the full sequences at the coarse and fine STS scales, respectively, reconstructs the solutions via transposed convolutions at corresponding resolutions separately, and then merges them to achieve multi-resolution solutions. This approach guarantees consistency with the coarse patch solution when $\gamma_{\mathrm{sts}} = 1.0$ (no refinement) and the fine patch solution when $\gamma_{\mathrm{sts}} = 0.0$ (refining all patches). Let $\boldsymbol{Z}'^{L}_{\mathrm{coarse}} = [\boldsymbol{z}^L_1, \boldsymbol{z}^L_2, \ldots, \boldsymbol{z}^L_{npx_1 \times npy_1 - N_{\mathrm{sts}}}] \in \mathbb{R}^{(npx_1 \times npy_1 - N_{\mathrm{sts}}) \times C_{\mathrm{emb}}}$ denote the coarse portion of the mixed-resolution attention output, and let $\boldsymbol{Z}^L_{\mathrm{sts},i} = [\boldsymbol{z}^L_1, \boldsymbol{z}^L_2, \ldots, \boldsymbol{z}^L_{p_{x_1}/p_{x_{\mathrm{sts}}} \times p_{y_1}/p_{y_{\mathrm{sts}}}}]_i$ $(i = 1, \ldots, N_{\mathrm{sts}})$ denote the STS portion. `Adap_Mix` performs the following operations:

1. Reconstruct the full coarse patches $\boldsymbol{Z}^L_{\mathrm{coarse}} \in \mathbb{R}^{npx_1 \times npy_1 \times C_{\mathrm{emb}}}$ via

$$\begin{aligned}
\boldsymbol{Z}^L_{\mathrm{coarse}}[\text{Kep-IDs}] &= \boldsymbol{Z}'^{L}_{\mathrm{coarse}}, \\
\boldsymbol{Z}^L_{\mathrm{coarse}}[\text{STS-IDs}] &= [\mathrm{Mean}(\boldsymbol{Z}^L_{\mathrm{sts},1}), \mathrm{Mean}(\boldsymbol{Z}^L_{\mathrm{sts},2}), \ldots, \mathrm{Mean}(\boldsymbol{Z}^L_{\mathrm{sts},N_{\mathrm{sts}}})],
\end{aligned} \tag{6}$$

   where Kep-IDs is the complementary indexing tensor to STS-IDs, representing all coarse patches kept in the sequence.

2. Reconstruct the full fine patches $\boldsymbol{Z}^L_{\mathrm{fine}} \in \mathbb{R}^{H/p_{x_{\mathrm{sts}}} \times W/p_{y_{\mathrm{sts}}} \times C_{\mathrm{emb}}}$ via

$$\begin{aligned}
\boldsymbol{Z}'^{L}_{\mathrm{fine}}[\text{STS-IDs}, :, :] &= [\boldsymbol{Z}^L_{\mathrm{sts},1}, \boldsymbol{Z}^L_{\mathrm{sts},2}, \ldots, \boldsymbol{Z}^L_{\mathrm{sts},N_{\mathrm{sts}}}], \\
\boldsymbol{Z}'^{L}_{\mathrm{fine}}[\text{Kep-IDs}, :, :] &= \mathrm{repeat}\left(\boldsymbol{Z}'^{L}_{\mathrm{coarse}}, p_{x_1}/p_{x_{\mathrm{sts}}} \times p_{y_1}/p_{y_{\mathrm{sts}}}\right), \\
\boldsymbol{Z}^L_{\mathrm{fine}} &= \mathrm{reshape}\left(\boldsymbol{Z}'^{L}_{\mathrm{fine}}\right).
\end{aligned} \tag{7}$$

   where $\boldsymbol{Z}'^{L}_{\mathrm{fine}} \in \mathbb{R}^{(npx_1 \times npy_1) \times (p_{x_1}/p_{x_{\mathrm{sts}}} \times p_{y_1}/p_{y_{\mathrm{sts}}}) \times C_{\mathrm{emb}}}$ is an intermediate supporting tensor.

3. Reconstruct solution fields $\hat{\boldsymbol{u}}_{t,\mathrm{coarse}} \in \mathbb{R}^{H \times W \times C}$ and $\hat{\boldsymbol{u}}_{t,\mathrm{fine}} \in \mathbb{R}^{H \times W \times C}$ from coarse patches and fine patches, respectively:

$$\hat{\boldsymbol{u}}_{t,\mathrm{coarse}} = \mathrm{ConvTranspose2d}_1(\boldsymbol{Z}^L_{\mathrm{coarse}}), \quad \hat{\boldsymbol{u}}_{t,\mathrm{fine}} = \mathrm{ConvTranspose2d}_2(\boldsymbol{Z}^L_{\mathrm{fine}}). \tag{8}$$

4. Fusion of solutions from step 3 to get the multi-resolution solution fields $\hat{\boldsymbol{u}}_t \in \mathbb{R}^{H \times W \times C}$ :

$$\hat{\boldsymbol{u}}_t[\text{Kep-IDs}] = \hat{\boldsymbol{u}}_{t,\mathrm{coarse}}[\text{Kep-IDs}], \quad \hat{\boldsymbol{u}}_t[\text{STS-IDs}] = \hat{\boldsymbol{u}}_{t,\mathrm{fine}}[\text{STS-IDs}]. \tag{9}$$

Among the two adaptive approaches, `Adap_Mul` is simpler to implement, requiring minimal code modifications, supports the AViT attention mechanism, and does not increase the maximum sequence lengths. In contrast, `Adap_Mix` produces relatively longer sequences and lacks AViT support but has the potential significant benefit of better capturing cross-scale correlations than the decoupled `Adap_Mul`. Furthermore, by varying $\gamma_{\text{sts}}$ from 1.0 to 0.0, `Adap_Mix` guarantees a smooth transition from the coarse patch solution at $[p_{x_1}, p_{y_1}]$ to the fine patch solution at $[p_{x_{\text{sts}}}, p_{y_{\text{sts}}}]$ (see Figure 5).

`Adap_Mix` and MSViT (Havtorn et al., 2023) are both mixed-scale tokenization methods, but they differ in that `Adap_Mix` refines patches directly based on the variance of input data, rather than relying on an auxiliary NN and potentially unknown prior distributions. As a result, it is easier to implement, more effective (see Section 4.2), and also more extensible, since the variance indicator can be swapped out by other physical indicators. Moreover, while MSViT's combination of coarse and refined tokens in attention is broadly analogous to `Adap_Mix`, they do not consider the potentially more computationally efficient `Adap_Mul`.

## 4 Experiments

We design three experiments to evaluate 1) the baseline performance of three spatiotemporal attention schemes (AViT, SViT, and ViT) using constant uniform tokenization, 2) the impact of adaptive tokenization on each spatiotemporal attention scheme, and 3) the effectiveness of pretrained models on two fine-tuning tasks that feature physics different from the pretraining data. In these experiments, we set $p_t = 1$ and $C_{\text{uni}} = C_{\text{emb}}/4$, and employ square patches (i.e., $p_x = p_y$, $p_{x_1} = p_{y_1}$, and $p_{x_{\text{sts}}} = p_{y_{\text{sts}}}$) by default.

### 4.1 Spatiotemporal attention schemes

We evaluate AViT, SViT, and ViT for three model sizes: Tiny (Ti), Small (S), and Base (B) with 3, 6, 12 heads and hidden dimension $C_{\text{emb}} = 192, 384,$ and $768$, respectively (Touvron et al., 2022), as shown in Table 1, on the colliding thermals dataset. In the same size category, AViT and SViT are about 30% larger than ViT due to the additional attention block. More details about the experiment are presented in Appendix A.2.

|      | Tiny  | Small | Base   |
|------|-------|-------|--------|
| AViT | 7.5M  | 29.9M | 119.3M |
| SViT | 7.6M  | 30.0M | 119.3M |
| ViT  | 5.8M  | 22.8M | 90.9M  |

Table 1: Number of model parameters in AViT, SViT, and ViT for three model sizes, Tiny, Small, and Base, detailed in Section 4.1. ViT results in about 30% fewer model parameters than AViT and SViT because the latter two require additional attention blocks.

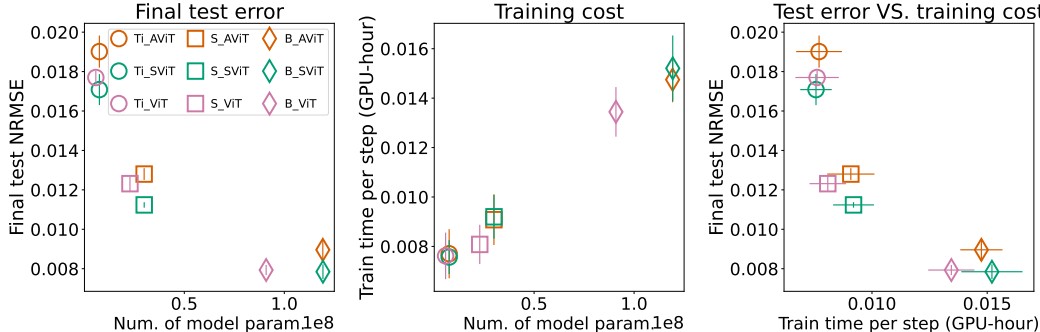

Figure 3: Learning efficiency of AViT, SViT, and ViT at three model sizes regarding final predictive error and training time cost: SViT and ViT are observed to be more expressive and computationally efficient than AViT in the experiment, as they require fewer model parameters and less training time to achieve the same test accuracy.

Figure 3 compares the final test error, defined as the normalized root-mean-square error (NRMSE), and the training time, represented as GPU hour per step, for the nine models. For the same size category, SViT (green) achieves the lowest error, followed by ViT (magenta), and then AViT (red-orange). In terms of training time, SViT takes longer than AViT, while ViT is the least expensive. ViT processes longer token sequences and hence is expected to have a higher single-unit attention cost, whereas AViT and SViT have multiple attention units with shorter token sequence lengths. The results reported in Figure 3 show that the ViT has the lowest cost, which implies that the number of attention blocks plays a more important role than the token sequence length in terms of training cost in this example. This observation is due to the fact that the spatiotemporal token sequence length ($16 \times 8 \times 8$) in this example is relatively short. We expect ViT to become more expensive than AViT and SViT when more refined or higher dimensional solutions are considered, in which longer token sequences are required.

In general, we find that SViTs and ViTs are more expressive and computationally efficient than AViTs, in that they achieve the lower predictive errors with fewer model parameters and less training time in relatively short token sequences. These baseline results provide several insights for our adaptive tokenization approach and experimental design in the subsequent section. First, as the primary drawback of SViT and ViT is the expected increase in compute required for longer sequence lengths, potential length reductions from adaptive tokenization would be more impactful for these two schemes. Moreover, such a reduction would be most significant computationally if it were to minimize any increases in the attention operations that scale quadratically with sequence lengths. Second, as the relative behavior of the three attention schemes is consistent across the three model sizes considered, we can select the smallest model for the adaptive tokenization experiments. Third, the lower expressiveness of the fully decoupled AViT suggests that the coupling between the coarse and refined tokens in an adaptive tokenization scheme may also have trade-offs between computational efficiency and expressiveness.

## 4.2 Adaptive tokenization

~~We start the evaluation of our adaptive tokenization methods in a single collision trajectory between two thermal bubbles.~~ Figure 4 compares the temperature contours of the true solution at $t = 590$ with the predicted solutions from Ti-SViT models at constant patch sizes, ps=$16 \times 16$ and ps=$32 \times 32$, and adaptive tokenization (`Adap_Mul` with $p_{x_1} = p_{y_1} = 32$, $p_{x_\mathrm{sts}} = p_{y_\mathrm{sts}} = 16$ , and $\gamma_\mathrm{sts} = 0.2$). The predicted solution from ps=$32 \times 32$ exhibits abrupt changes with clear edges for the local structures inside the patches, while the finer resolution model at ps=$16 \times 16$ captures smoother, finer structures but requires many more patches. In contrast, our adaptive tokenization methods capture smooth, fine structures comparable to ps=$16 \times 16$ while requiring much shorter sequences.

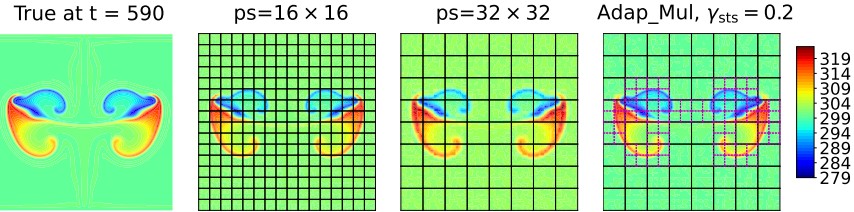

Figure 4: Predicted temperature contours at $t = 590$ from Ti-SViT models with constant patch sizes ps=$16 \times 16$ and ps=$32 \times 32$ and adaptive tokenization (`Adap_Mul` with $p_{x_1} = p_{y_1} = 32$, $p_{x_\mathrm{sts}} = p_{y_\mathrm{sts}} = 16$, and $\gamma_\mathrm{sts} = 0.2$). `Adap_Mul` predicts smoother, finer local structures that are overlooked in ps=$32 \times 32$, similar to the more expensive ps=$16 \times 16$.

Next, we evaluate the two adaptive tokenization methods, `Adap_Mix` and `Adap_Mul`, across the three attention schemes. We begin with a single collision trajectory between two thermal bubbles for efficient hyperparameter sweeps, then compare `Adap_Mix` with MSViT using a gating-NN-controlled mixed-scale tokenization, and finally demonstrate the generalizability of our adaptive tokenization methods on 576 trajectories with varying bubble locations and intensities.

**Adap_Mix in ViT and SViT** Adap_Mix with ($p_{x_1}$, $p_{x_{\text{sts}}}$, $\gamma_{\text{sts}}$) is designed to ensure convergence in $\gamma_{\text{sts}}$ values. When $\gamma_{\text{sts}} \to 1$, no refinement is conducted and the output is converged to ps=$p_{x_1} \times p_{x_1}$. Conversely, when $\gamma_{\text{sts}} \to 0$, all patches are refined and the output is converged to ps=$p_{x_{\text{sts}}} \times p_{x_{\text{sts}}}$. To examine such convergence behavior, we conduct a set of runs with varying ($p_{x_1}$, $p_{x_{\text{sts}}}$, $\gamma_{\text{sts}}$) values, together with runs at constant patch sizes, ps=$32 \times 32$, ps=$16 \times 16$, and ps=$8 \times 8$. Figure 5 shows the final NRMSE test loss versus the average sequence length of patches per time step, $L_{\text{avg,mix}}$. For a given trajectory of spatiotemporal solutions with $T$ steps, the average sequence length is defined as

$$L_{\text{avg,mix}} = \frac{1}{T} \sum_{t=1}^{T} L_t = \frac{1}{T} \sum_{t=1}^{T} \left[ (npx_1 \cdot npy_1 - N_{\text{sts},t}) + N_{\text{sts},t} \cdot \left( \frac{p_{x_1}}{p_{x_{\text{sts}}}} \cdot \frac{p_{y_1}}{p_{y_{\text{sts}}}} \right) \right], \tag{10}$$

where $L_t$ represents the length of the mixed patch sequence and $N_{\text{sts},t}$ is the number of patches selected for refinement based on Equation (4) at time $t$. Clearly, the predictive error of Adap_Mix evolves from the corresponding coarse patch results of ps=$p_{x_1} \times p_{x_1}$ to ps=$p_{x_{\text{sts}}} \times p_{x_{\text{sts}}}$ when $\gamma_{\text{sts}}$ varying from 1.0 to 0.0, in two settings ($p_{x_1} = 32, p_{x_{\text{sts}}} = 16$) and ($p_{x_1} = 16, p_{x_{\text{sts}}} = 8$) and for both ViT and SViT. More interestingly, Adap_Mix at some $\gamma_{\text{sts}}$ values even achieves a lower predictive error than the fine patch case ps=$p_{x_{\text{sts}}} \times p_{x_{\text{sts}}}$ with a much shorter sequence length (e.g., with $2\times$ reduction), clearly indicating the advantages of the adaptive tokenization approach.

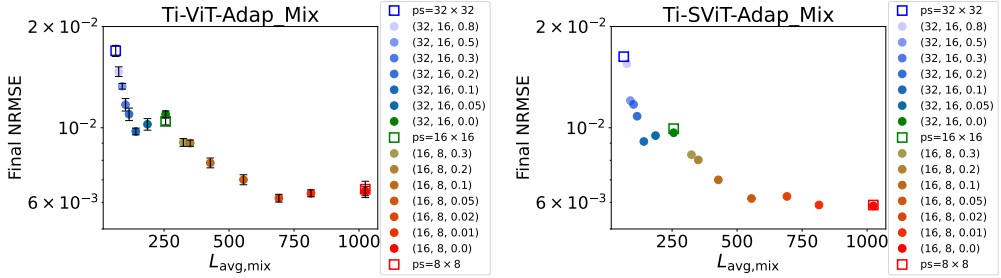

Figure 5: Final NRMSE for Tiny ViT (left) and SViT (right) with adaptive tokenization — Adap_Mix with hyperparameters ($p_{x_1}$, $p_{x_{\text{sts}}}$, $\gamma_{\text{sts}}$)— and constant patch sizes against average sequence length, $L_{\text{avg,mix}}$ (Equation (10)). Error bars, representing standard deviations from 3 runs, are shown for ViT. Adap_Mix with $\gamma_{\text{sts}}$ varying from 1.0 to 0.0 shows a clear convergent transition from the coarse constant patch size ps=$p_{x_1} \times p_{y_1}$ to the fine constant patch size ps=$p_{x_{\text{sts}}} \times p_{y_{\text{sts}}}$. More interestingly, Adap_Mix is shown to achieve lower prediction errors than the more expensive ps=$p_{x_{\text{sts}}} \times p_{y_{\text{sts}}}$ cases despite requiring only half of the average sequence length. Corresponding numbers are reported in Table A1.

**Adap_Mul in ViT, SViT, and AViT** In contrast to Adap_Mix, which combines the coarse and locally refined patches into a hybrid sequence that is fed into attention blocks together, Adap_Mul treats the two, i.e., $\boldsymbol{Z}_{\text{coarse}}^{L} = \left[ \boldsymbol{z}_1^L, \boldsymbol{z}_2^L, \ldots, \boldsymbol{z}_{npx_1 \times npy_1}^L \right]$ and $\boldsymbol{Z}_{\text{sts},i}^{L} = \left[ \boldsymbol{z}_{\text{sts},1}^L, \boldsymbol{z}_{\text{sts},2}^L, \ldots, \boldsymbol{z}_{sts,p_{x_1}/p_{x_{\text{sts}}} \times p_{y_1}/p_{y_{\text{sts}}}}^L \right]_i$ ($i = 1, \ldots, N_{\text{sts}}$), separately. Adap_Mul maintains this separation for both attention and solution reconstruction and views the reconstructed solutions from the refined patches as a local STS correction. The computing cost scales either linearly for MLP or quadratically for attention with sequence length in various model components. To represent the cost, we define the linear and quadratic indices for ViT and SViT as in

$$L_{\text{lin}} = \frac{1}{T} \sum_{t=1}^{T} L_t = \frac{1}{T} \sum_{t=1}^{T} \left[ npx_1 \cdot npy_1 + N_{\text{sts},t} \cdot \left( \frac{p_{x_1}}{p_{x_{\text{sts}}}} \cdot \frac{p_{y_1}}{p_{y_{\text{sts}}}} \right) \right],$$

$$L_{\text{quad}} = \frac{1}{T} \sum_{t=1}^{T} L_{\text{quad},t} = \frac{1}{T} \sum_{t=1}^{T} \left[ (npx_1 \cdot npy_1)^2 + N_{\text{sts},t} \cdot \left( \frac{p_{x_1}}{p_{x_{\text{sts}}}} \cdot \frac{p_{y_1}}{p_{y_{\text{sts}}}} \right)^2 \right]. \tag{11}$$

For AViT, the index $L_{\text{quad}}$ needs to be adjusted as

$$L_{\text{quad}} = \frac{1}{T} \sum_{t=1}^{T} \left[ (npx_1^2 \cdot npy_1 + npx_1 \cdot npy_1^2) + N_{\text{sts},t} \cdot \left[ \left( \frac{p_{x_1}}{p_{x_{\text{sts}}}} \right)^2 \cdot \frac{p_{y_1}}{p_{y_{\text{sts}}}} + \frac{p_{x_1}}{p_{x_{\text{sts}}}} \cdot \left( \frac{p_{y_1}}{p_{y_{\text{sts}}}} \right)^2 \right] \right]. \tag{12}$$

Figure 6 shows the final test NRMSE against the cost indices $L_{\text{lin}}$ (left) and $L_{\text{quad}}$ (right) of `Adap_Mul` in ViT (top) and SViT (bottom) at varying values of $(p_{x_1}, p_{x_{\text{sts}}}, \gamma_{\text{sts}})$. The results on the left panel confirm that, as $\gamma_{\text{sts}}$ decreases, `Adap_Mul` refines more aggressively, resulting in lower NRMSE while increasing the linear cost. However, the plots on the right panel show that `Adap_Mul` achieves much lower NRMSE at minimal additional quadratic cost. This observation suggests that `Adap_Mul` leads to preferable performance on problems with attention on long sequence, where the quadratic cost is dominant (Touvron et al., 2022). Figure 7 presents analogous results for `Adap_Mul` in AViT. Compared with ViT and SViT, the accuracy improvement in AViT is less pronounced in our experiments with $(p_{x_1}, p_{x_{\text{sts}}}) = (32, 16)$ and $(16, 8)$ (top), possibly due to the extremely short sequence lengths of 2 in AViT. Notably, significant accuracy gains are observed when $p_{x_1} = 32$ and $p_{x_{\text{sts}}}$ reduced from 16 to 8 (bottom).

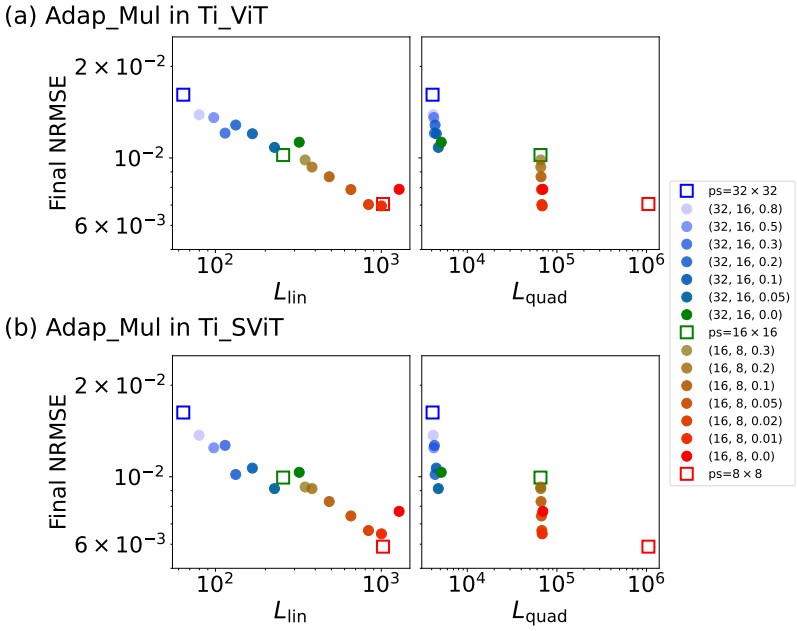

Figure 6: Final NRMSE for Tiny ViT (a) and SViT (b) with adaptive tokenization — `Adap_Mul` with hyperparameters $(p_{x_1}, p_{x_{\text{sts}}}, \gamma_{\text{sts}})$ — and constant patch sizes against linear cost estimation index $L_{\text{lin}}$ (left) and quadratic cost estimation index $L_{\text{quad}}$ (right) (Equation (11)). As $\gamma_{\text{sts}}$ reduces from 1.0 to 0.0, `Adap_Mul` leads to decreasing NRMSE with increasing linear cost increases, whereas the increment of quadratic cost remains negligible. Corresponding numbers are reported in Table A2.

Comparing the two approaches for adaptive tokenization, we find that `Adap_Mix` provides better predictive accuracy, likely due to considering cross-scale correlations, and guarantees convergence toward the solution with uniformly refined tokens. In contrast, `Adap_Mul` is dramatically more cost effective for attention operations with quadratic complexity and easier to implement than `Adap_Mix`. AViT does not interact well with adaptive tokenization approaches when the STS sequence length $p_{x_1}/p_{x_{\text{sts}}}$ is too short. Decreasing $\gamma_{sts}$ values generally improves accuracy at a case-dependent computing cost increase. For `Adap_Mul`, lowering $\gamma_{sts}$ incurs negligible extra attention cost (the dominate cost for long sequence) but yields substantial accuracy gain; accordingly, we would recommend $\gamma_{sts} \to 0$. For `Adap_Mix`, smaller $\gamma_{sts}$ values generate better results but also faster growing sequence length (while still providing significantly speedup); we would suggest the value based on computing resources: identifying the sequence length from memory and compute hour limitations and then working backward to select $\gamma_{sts}$ based on data and the set of patch sizes.

**`Adap_Mix` VS. MSViT** To compare `Adap_Mix` with MSViT, we implemented the gate NN controlled adaptive tokenization in MSViT (Havtorn et al., 2023) and ran six configurations at three different target gate sparsity $g^\star$ values ($g^\star = [0.1, 0.5, 0.9]$, where 0 represents no refinement and 1 represents full refinement) and two sets of patch size values, i.e., $(p_{x_1}, p_{x_{\text{sts}}}) = [(32, 16), (16, 8)]$. We used the GBaS loss with a

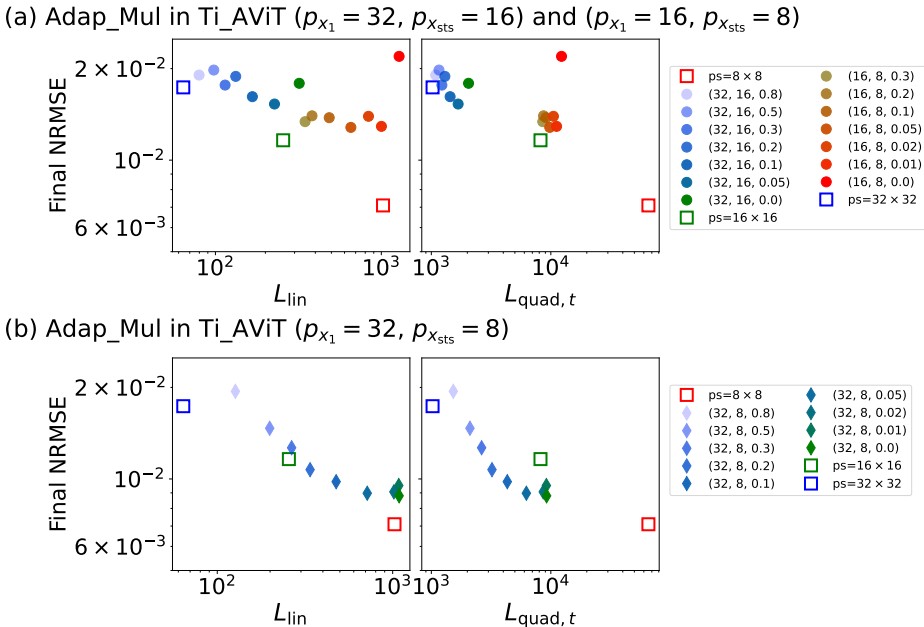

Figure 7: Final NRMSE for Tiny AViT with adaptive tokenization — `Adap_Mul` with hyperparameters $(p_{x_1}, p_{x_{sts}}, \gamma_{sts})$ — and constant patch sizes against linear cost estimation index $L_{lin}$ (Equation (11)) (left) and quadratic cost estimation index $L_{quad}$ (Equation (12)) (right). The axial attention in AViT results in short attention sequence, and thus the advantage of `Adap_Mul` is not observed in (a) when $(p_{x_1}, p_{x_{sts}}) = (32, 16)$ and $(16, 8)$. In (b), $(p_{x_1}, p_{x_{sts}})$ is chosen to be $(32, 8)$, which increases the ratio between $p_{x_1}$ and $p_{x_{sts}}$ from 2 to 4. In this case, `Adap_Mul` gives substantial NRMSE reduction at a relatively low quadratic cost.

Gaussian hyperprior and a Bernoulli prior following (Havtorn et al., 2023). Figure 8 presents the final test NRMSE against the average sequence length of patches per time step, $L_{avg,mix}$ (Equation (10)) for MSViT, in comparison with `Adap_Mix` and constant patch sizes from the left subplot of Figure 5. The results show that `Adap_Mix` (circles) is more efficient than MSViT (stars), achieving better accuracy with shorter sequence lengths. This efficiency is due to `Adap_Mix` directly selecting patch refinement based on input data, whereas MSViT relies on a gating NN jointly trained with the model. The gate begins with an initial condition that refines all patches (i.e., high initial $L_{avg,mix}$ values). During training, MSViT operates with longer sequence length until the gating NN converges, which occurs after the convergence of the hyperprior distribution. We omit a direct comparison between `Adap_Mul` and MSViT for the same reason that `Adap_Mul` and `Adap_Mix` are plotted separately; they are fundamentally two different approaches—`Adap_Mul` treats tokens at different scales separately in a much more computational efficient way, while MSViT shares more similarities with `Adap_Mix`.

**`Adap_Mix` and `Adap_Mul` in multiple trajectories** To evaluate the generalizability, we employed 576 collision trajectories—512 for training and 64 for testing—each spanning 1001 time steps (see Appendix A.1.1). Due to computational cost, we focused on $\gamma_{sts} \in [0.1, 0.2, 0.4, 0.8])$, drawn from the single-trajectory experiments in Figures 5, 6, and 7. Figure 9 shows the final NRMSE of `Adap_Mix` in ViT (left) and SViT (right) against the average sequence length, alongside constant patch-size baselines. As $\gamma$ decreases from 0.8 to 0.1, NRMSE steadily decreases, consistent with Figure 5. Figure 10 presents analogous results of `Adap_Mul`, again matching the single-trajectory results in Figures 6 and 7. These findings confirm that both adaptive algorithms generalize effectively to complex, spatially and temporally inhomogeneous trajectories.

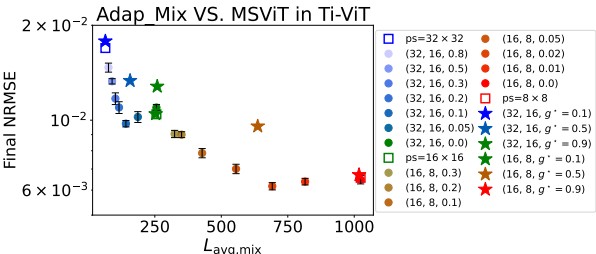

Figure 8: Final NRMSE for `Adap_Mix` and MSViT (Havtorn et al., 2023) in Tiny ViT against average sequence length, $L_{\text{avg,mix}}$ (Equation (10)). The results of `Adap_Mix` and constant patch sizes are the same as in the left subplot of Figure 5. Star symbols represent MSViT results. The sequence length is collected during training, for which the gate loss history is shown in A7. `Adap_Mix` is observed to be more efficient than MSViT, achieving higher accuracy with shorter sequences. This is because `Adap_Mix` refines patches directly based on input data, while MSViT uses a jointly trained gating NN that initially refines all patches (hence high $L_{\text{avg,mix}}$ values). Thus, MSViT runs with longer sequences during training until the gate converges, which happens after the hyperprior distribution stabilizes. Corresponding MSViT numbers are reported in Table A3.

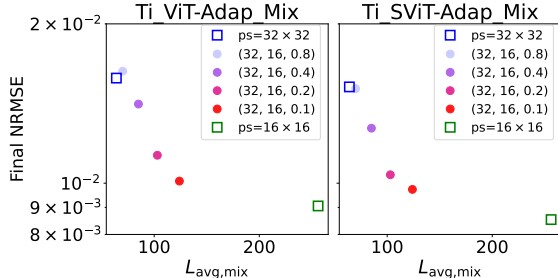

Figure 9: Final NRMSE for Tiny ViT (left) and SViT (right) against average sequence length $L_{\text{avg,mix}}$ (Equation (10)), comparing `Adap_Mix` with hyperparameters ($p_{x_1}$, $p_{x_{\text{sts}}}$, $\gamma_{\text{sts}}$) and constant-patch-size baselines, trained on multiple collision trajectories. The reduction in error as $\gamma_{\text{sts}}$ decreases is consistent with the single-trajectory results in Figure 5.

### 4.3 Effectiveness of pretraining in colliding thermals and MHD fine-tuning tasks

We examine the transferrability of pretrained models to fine-tuning systems with distinct physics and different set of variables, as in Table A4. Specifically, we aim to address three broad questions: (1) Is pretraining effective when the downstream tasks have a distinct set of physical variables? (2) How does limited fine-tuning of non-attention blocks compare to full fine-tuning? and (3) How does the amount of fine-tuning data affect convergence? To address these three questions, we design a sets of experiments, starting from models pretrained on PDEBench or randomly initialized models ("*_INIT"), and fine-tune them on colliding thermals and MHD datasets with distinct physical variables. For fine-tuning each model, we either allow all model parameters to be tunable ("ALL") or freeze the attention blocks and limit training to the preprocessor, the tokenization module, and the postprocessor ("PREPOST"). Finally, for each initial model and fine-tuning configuration, we train four models with increasing amounts of fine-tuning data.

For the colliding thermals dataset, Figure 11 compares the test loss with full and limited fine-tuning using pretrained and randomly initialized models. The different training data sizes ranging from one set of colliding thermals time-trajectory to 24 sets of trajectories. The fine-tuning task is to predict the solution of the physical system at a lead time of $t_{\text{lead}}$ uniformly sampled between 1 and 50 steps. An example of the true and predicted solutions in these four training configurations is illustrated in Figure 12.

For the limited fine-tuning test with the colliding thermals dataset, the pretrained models achieve significantly lower error than starting from scratch with randomly initialized parameters. Moreover, while this advantage

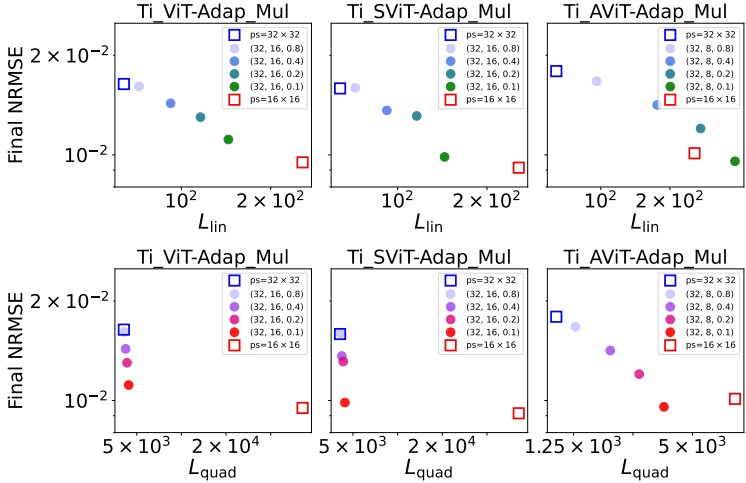

Figure 10: Final NRMSE for Tiny ViT (left), SViT (middle), and AViT (right) against the linear cost estimation index $L_{\text{lin}}$ (Equation (11), top row) and the quadratic cost estimation index $L_{\text{quad}}$ (Equation (12), bottom row). Symbols compare `Adap_Mul` adaptive tokenization with hyperparameters $(p_{x_1}, p_{x_{\text{sts}}}, \gamma_{\text{sts}})$ and constant-patch-size baselines, trained on multiple collision trajectories. The changing trend in error and cost as $\gamma_{\text{sts}}$ decreases again mirrows the single-trajectory results Figures 6 and 7, demonstrating the generalizability of the adaptive algorithms to spatially and temporally inhomogeneous trajectories.

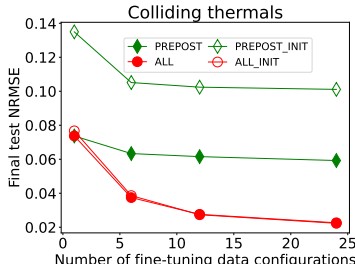

Figure 11: NRMSE for test set at different training data sizes in fine-tuning of colliding thermals at a maximum lead time of 50 steps, with full ("ALL") and limited ("PREPOST") fine-tuning using pretrained and randomly initialized models ("*_INIT").

persists as the number of fine-tuning data increases, it is most pronounced in the low data configuration of learning from a single trajectory. Indeed, we find that limited fine-tuning with the pretrained models generalizes well even when learning from one trajectory, seeing only moderate improvements when run on the largest dataset size considered. Overall, the lower converged error from pretrained models suggests the frozen attention blocks clearly learned transferable knowledge during pretraining. For full fine-tuning, the accuracy is much better than limited fine-tuning as a result of the model being more expressive. The difference between the pretrained and randomly initialized models is much lower, being minor in the case of a single data configuration during training and vanishing as the amount of data increases.

For the MHD dataset, Figure 13 shows the final test NRMSE errors in lid-driven cavity flows after fine-tuning against data sizes when starting from pretrained and randomly initialized models for limited and full fine-tuning. The training dataset sizes used for fine-tuning range from 1 to 12 simulation configurations, with each configuration including approximately 1900 samples. The fine-tuning task is to predict the flow solution at a lead time of $t_{\text{lead}}$ uniformly sampled between 1 and 100 steps. Contour plots from the true solution and the predicted solution from each training configuration are depicted in Figure 14.

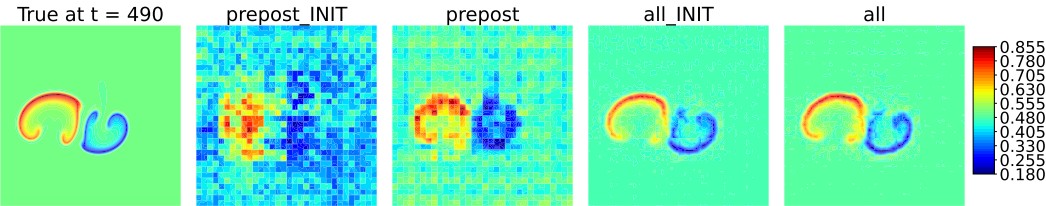

Figure 12: Temperature contours of true solution vs predicted solutions from four fine-tuned models (on 12 trajectories) at $t = 490$ from Ti-SViT models for a lead time of 40 in the collision of two thermal bubbles.

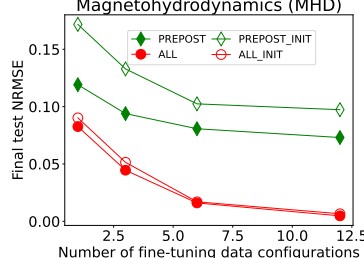

Figure 13: NRMSE for test set at different training data sizes in fine-tuning of lid-driven cavity MHD flows dataset at a maximum lead time of 100 steps, with full ("ALL") and limited ("PREPOST") fine-tuning using pretrained and randomly initialized models ("*_INIT").

Overall, the fine-tuning performance is a result of model expressibility, training data size, and the similarity between training and testing tasks. As with the colliding thermals dataset, pretrained models outperformed the randomly initialized models for both full and limited fine-tuning runs. However, the reduced expressibility of the limited fine-tuning configuration consistently shows an accuracy gap, even with more training data, as they cannot fully represent the data complexity. In contrast, full fine-tuning leads to more expressive models that can capture all training data information when trained on limited data but often show high test errors; as more training data is provided, they generalize well and lead to a convergent improved test error. In our fine-tuning, the randomly initialized models perform well in testing even with a single data configuration, likely due to the similarity between training and testing tasks. Future work will explore more challenging scenarios with increased heterogeneity within the fine-tuning data.

While studies like McCabe et al. (2023) have demonstrated impressive outperformance from fine-tuning of pretrained models versus randomly initialized models, these fine-tuning tests were performed on data that, while distinct, was fully governed by physical equations and characterized by physical variables that were represented in the training data. Yet for a model that aims to be foundational for multiphysical systems, we argue that assessing model performance in more realistic settings, where equations like Navier-Stokes are coupled with those from other domains of physics, is a more informative test of the effectiveness of pretraining. Accordingly, we assess fine-tuning performance on physical systems that incorporate fluid flows, which are well-represented in PDEBench, with thermodynamics and electromagnetism, which are not. As reasonably anticipated, we find that advantages of pretraining are reduced in this more complex setting.

## 5 Conclusion

In this paper, we make three contributions that will advance the development of foundation models for multiscale physical systems. First, we find that while some data efficiency is lost in a fully decoupled spatiotemporal attention scheme such as AViT, SViT provides an intriguing balance of computational and data efficiency versus the standard ViT approach. Yet using SViT alone does not sufficiently address the computational challenges associated with attention for high spatial resolutions. Second, we instead suggest that our adaptive tokenization scheme provides a promising approach for working with high resolution data. This sort of adaptivity has the potential to be both flexible and expressive enough to deal with the dynamic

Figure 14: Contours of true horizontal magnetic field values $B_x$ vs predicted solutions from four fine-tuned models (on 12 trajectories) at $t = 1400$ from Ti-SViT models for a lead time of 80 in lid-driven cavity MHD flows.

and sparse nature of the multiscale features in physical data. Third, we suggest an alternative path to evaluate foundation models for multiscale physical systems that focuses on fine-tuning problems involving out-of-distribution physics governed by different equations with distinct sets of physical variables. In two such settings, colliding thermals and magnetohydrodynamics, we find that while pretraining does provide an advantage, its impact is much more muted compared to fine-tuning on the same set of variables, suggesting additional effort is required to obtain truly foundational models in this space.

This work focused on demonstrating the effectiveness of our approach in 2D systems. Future directions include moving toward real-world applications with larger model sizes using high-resolution 3D data. In these directions, the computational challenges considered here will only increase, especially the substantially longer sequence lengths associated with 3D systems. Both adaptive tokenization methods introduced here, in combination with compatible spatiotemporal attention schemes like SViT, can play a significant role in moderating the additional computational costs in these regimes. However, as sequence lengths move into the millions or more, additional techniques such as parallelizing a sequence across multiple GPUs will be necessary as well.

## Broader Impact Statement

This study presents fundamental algorithms designed to advance transformer-based spatiotemporal foundation models. These models are trained and tested on both open-source and in-house 2D simulation data from PDE systems, representing basic SciML research. The in-house data will be released upon paper publication. At this proof-of-concept stage, the models pose minimal risk and have the negligible negative societal impacts.

Our long-term objective is to develop Multiscale AdapTivE trustworthY (MATEY) transformer-based models for real-world multiscale physical systems such as propulsion, energy generation, and earth sciences. Developing models for such applications makes trustworthiness — specifically, the model's accuracy and reliability, robustness, and resilience (ARRR) — a primary concern. While this paper demonstrates an early proof-of-concept demonstration toward that goal, the models are not yet deployed in practical applications. To facilitate transparency and reproducibility, and to help reduce potential bias in data and implementations, we will make our codebase, datasets, and experimental settings open source upon publication.

## Acknowledgments

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

# A Appendix

## A.1 Datasets

Three datasets were used in the work: PDEBench (Takamoto et al., 2022), colliding thermals (Norman, 2024), and lid-driven cavity MHD flows.

- PDEBench (`https://github.com/pdebench/PDEBench`) consists of diverse 1D, 2D, and 3D diverse benchmark datasets. We used the 2D cases – incompressible flows, compressible flows, turbulent flows, reaction diffusion, and shallow water – for model pretraining in Section 4.3. The governing equations are summarized below.

  - Shallow water equations [swe]:
  $$\partial_t h + \nabla \cdot (h\boldsymbol{v}) = 0,$$
  $$\partial_t (h\boldsymbol{v}) + \nabla \cdot \left( \frac{1}{2} h\boldsymbol{v}^2 + \frac{1}{2} g_r h^2 \right) = -g_r h \nabla b$$

  - Diffusion-reaction equations [diffre2d]:
  $$\partial_t \boldsymbol{c} = \boldsymbol{D}\nabla^2 \boldsymbol{c} + \boldsymbol{R}(\boldsymbol{c}),.$$

  where $\xi$ and $\phi$ in $\boldsymbol{c} = [\xi, \phi]$ are the activator and the inhibitor, respectively.

  - Incompressible NS [Incomp]:
  $$\nabla \cdot \boldsymbol{v} = 0,$$
  $$\rho \left( \partial_t \boldsymbol{v} + \boldsymbol{v} \cdot \nabla \boldsymbol{v} \right) = -\nabla p + \eta \nabla^2 \boldsymbol{v} + \boldsymbol{f}$$

  - Compressible NS [compNS] with random and turbulent initial conditions:
  $$\partial_t \rho + \nabla \cdot (\rho \boldsymbol{v}) = 0,$$
  $$\rho \left( \partial_t \boldsymbol{v} + \boldsymbol{v} \cdot \nabla \boldsymbol{v} \right) = -\nabla p + \eta \nabla^2 \boldsymbol{v} + (\zeta + \eta/3)\nabla(\nabla \cdot \boldsymbol{v})$$
  $$\partial_t \left[ \epsilon + \frac{\rho \boldsymbol{v}^2}{2} \right] + \nabla \cdot \left[ \left( \epsilon + p + \frac{\rho \boldsymbol{v}^2}{2} \right) \boldsymbol{v} - \boldsymbol{v} \cdot \boldsymbol{\sigma}' \right] = 0$$

  with $\epsilon = p/\Gamma - 1$ and $\Gamma = 5/3$.

  For more details on these cases and equations, users are referred to(Takamoto et al., 2022).

- The colliding thermals dataset was generated for our work, and the details will be presented in Section A.1.1. It was used in the experiments in Sections 4.1 and 4.2, and also as one of the two fine-tuning cases in Section 4.3.

- Lid-driven cavity MHD dataset was also generated in our work, and it was used as the other fine-tuning case in Section 4.3. We will present the details in Section A.1.2.

### A.1.1 Colliding thermals

Thermal collision datasets contains multiple time history trajectories of the mixing of two bubbles-one cold bubble at the top colliding with a warm bubble at the bottom. Details about the governing equations can be found in Norman (2024). These trajectories start from different initial temperature conditions as

$$T_0(x, z) = 300.0 + T_{10}(x, z) + T_{20}(x, z), \tag{13}$$

with one hot $T_{10}$ and cold $T_{20}$ thermals being

$$T_{10}(x, z) = \begin{cases} Tc_1 \cos\left(\frac{\pi}{2} d_1(x, z)\right)^2, & \text{if } d_1(x, z) \leqslant 1 \\ 0, & \text{otherwise} \end{cases} \tag{14}$$

and

$$T_{20}(x,z) = \begin{cases} -Tc_2 \cos\left(\frac{\pi}{2}d_2(x,z)\right)^2, & \text{if } d_2(x,z) \leqslant 1 \\ 0, & \text{otherwise} \end{cases} \tag{15}$$

where $Tc_i$ is the center temperature amplitude and $d_i(x,z) = \sqrt{\frac{(x-xc_i)^2}{rx_i^2} + \frac{(z-zc_i)^2}{rz_i^2}}$ is the distance from thermal center $(xc_i, zc_i)$ for $i = 1, 2$. The thermals are elliptical in shape with the radius, $rx_i$ and $rz_i$, in x and z directions, respectively.

**Configurations**   We sample 4096 configurations with the thermals $(i = 1, 2)$ at different locations following uniform distribution,

$$xc_i \sim U[0.2L, 0.8L], \ zc_1 \sim U[0.2L, 0.3L], \text{ and } zc_2 \sim U[0.7L, 0.8L], \tag{16}$$

with different elliptical shapes also following uniform distribution,

$$rx_i \sim U[0.1L, 0.2L] \text{ and } rz_i \sim U[0.1L, 0.2L], \tag{17}$$

and with temperature amplitudes equally sampled from,

$$Tc_i \sim C\{10, 15, 20, 25\}. \tag{18}$$

The equations are solved by using a finite volume method with $nx = 256, ny = 256$ grid points in x and z directions, respectively. The simulations are advanced in time for 500 seconds and solutions are saved every 0.5 second. In total, we have 4096 trajectories, each with data at size $(nt = 1001, nx = 256, ny = 256)$.

### A.1.2   Lid-driven cavity magnetohydrodynamics (MHD) flows

The MHD dataset contains solution trajectories from initial conditions to steady states for a benchmark lid-driven cavity MHD flow problem in two dimensions with varying configurations. The MHD flow is governed by an incompressible Navier-Stokes equation with Lorentz force coupled with an induction equation with divergence cleaning. The detail formulation of the governing equations and problem setting for the lid-driven MHD cavity problem are given in Fambri et al. (2023).

**Configurations**   In this dataset, we include solution trajectories of the lid-driven cavity problem at three magnetic Reynolds numbers $Re_m = 100$, 200, and 500, each with ten external horizontal magnetic field magnitude $B_x = 0.05, 0.10, \ldots, 0.50$. This gives 30 different problem configurations. For each problem configuration, the fluid velocity field $\mathbf{v}$ and the magnetic field $\mathbf{B}$ are recorded on a 128×128 uniform spatial mesh for 2,000 time steps.

## A.2   More on spatiotemporal attentions and adaptive tokenization

**Training setting**   We randomly sampled a subset with 512 trajectories for training and 64 trajectories for testing for the results in Sections 4.1 and 4.2. During training, we use the `AdamW` optimizer with a learning rate equal to $10^{-4}$. Batch size was set to be 128 and accumulate gradient step was set to be 1. Models were trained for 20,000 steps. For cases with constant patch size, the value was set to be $32 \times 32$. Cases with a single trajectory was trained for 15000 steps with batch size 40 for faster hyperparameter sweeps. In MSViT cases, we used a Gaussian hyperprior distribution with standard deviation being 0.1 and a relaxed Bernolli prior distribution with temperature being 0.3. The loss weight hyperparameter $\lambda$ was set to be 10.

For the experiment on spatotemporal attention schemes in Section 4.1, we ran 9 cases with AViT, SViT, and ViT attention blocks at three sizes (Ti, S, and B). Figure A1 shows the loss history during training of the models for both training and test sets, and Figure A2 shows the training time cost.

For the experiment on adaptive tokenization in Section 4.2, Figures A3, A4, and A5 show the training losses of all models in a single colliding thermal trajectory for Figures 5, 6, and 7, respectively. Figures A6 and A7 show the training losses and gate losses, respectively, of all MSViT models in Figure 8. Figures A8 and A9 plot the training losses of all models in multiple colliding trajectories for Figures 9 and 10, respectively.

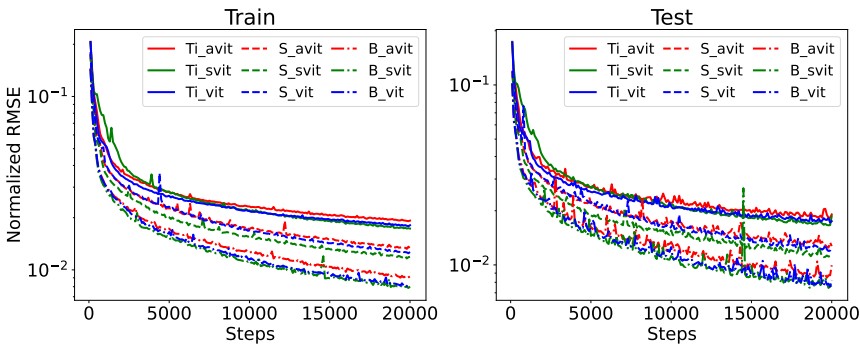

Figure A1: Loss history of three spatiotemporal attention schemes at three model sizes during training

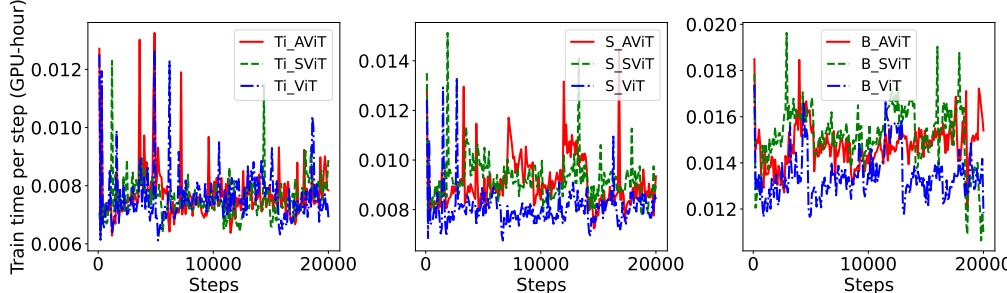

Figure A2: Training time per step of three spatiotemporal attention schemes at three model sizes

### A.3 Pretraining and fine-tuning

#### A.3.1 Pretraining

Five 2D datasets from PDEBench Takamoto et al. (2022) were used for pretraining, including shallow water, diffusion reaction, incompressible flows, compressible flows, and turbulent flows. The details of these datasets including physical variables, spatiotemporal resolutions, and number of trajectories are summarized in Table A4.

During training, we used the `AdamW` optimizer with `DAdaptAdam` for learning rate scheduling. Batch size was set to be 1472 and patch size was $32 \times 32$. Training/testing/validating split was 0.8/0.1/0.1. Gradient accumulation was set to be 1. We trained the model for 30,000 steps to predict the next step solution given a history of $T = 16$.

#### A.3.2 Fine-tuning

For fine-tuning, we evaluate the transferrability of pretrained models to systems with distinct physics and different sets of variables. Table A4 summarizes the two fine-tuning cases: colliding thermals and lid-driven cavity MHD flows. In the two cases, pretrained models were fine-tuned to predict the solution at a future time $t + t_{\text{lead}}$ given a history of solutions from $t - T + 1$ to $t$. In our experiments, $T$ was set to be 10 while $t_{\text{lead}}$ was set to 50 for the colliding thermals and 100 for the lid-driven cavity MHD flows. The fine-tuned models were evaluated on a held-out test set for all runs in each case. We used the `AdamW` optimizer with a learning rate equal to $10^{-4}$. Batch size was set to be 256. Models were fine-tuned for 600 epochs for colliding thermals and 1000 epochs for lid=drive cavity MHD flows.

**Colliding thermals** We sampled 1, 6, 12, and 24 trajectories for training. The results in Section 4.3 are shown for a fixed test set with 24 trajectories.

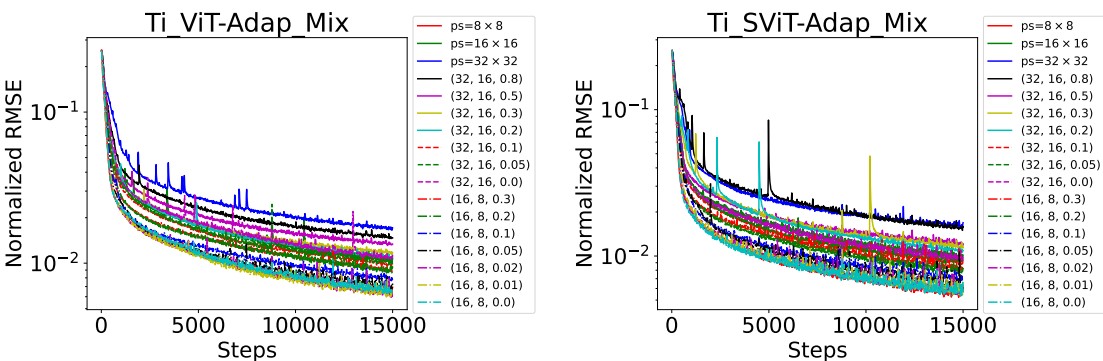

Figure A3: Comparison of training loss histories of models with adaptive tokenization Adap_Mix and constant patch sizes (ps=32 × 32, ps=16 × 16, and ps=8 × 8) for the two spatiotemporal attention schemes (ViT and SViT) in a single colliding thermals trajectory.

| Model | Metric | ps=8×8 | ps=16×16 | ps=32×32 | (32, 16, 0.8) | (32, 16, 0.5) | (32, 16, 0.3) | (32, 16, 0.2) | (32, 16, 0.1) | (32, 16, 0.05) | (32, 16, 0.0) | (16, 8, 0.3) | (16, 8, 0.2) | (16, 8, 0.1) | (16, 8, 0.05) | (16, 8, 0.02) | (16, 8, 0.01) | (16, 8, 0.0) |
|---|---|---|---|---|---|---|---|---|---|---|---|---|---|---|---|---|---|---|
| Ti-ViT-Adap_Mix | $L_{\mathrm{avg,mix}}$ | 1024 | 256 | 64 | 75 | 89 | 101 | 115 | 141 | 186 | 256 | 324 | 350 | 427 | 555 | 691 | 815 | 1024 |
| | NRMSE | 6.57e-03 | 1.04e-02 | 1.69e-02 | 1.47e-02 | 1.33e-02 | 1.17e-02 | 1.10e-02 | 9.75e-03 | 1.02e-02 | 1.10e-02 | 9.04e-03 | 9.00e-03 | 7.86e-03 | 7.01e-03 | 6.17e-03 | 6.39e-03 | 6.48e-03 |
| Ti-SViT-Adap_Mix | $L_{\mathrm{avg,mix}}$ | 1024 | 256 | 64 | 75 | 89 | 101 | 115 | 141 | 186 | 256 | 324 | 350 | 427 | 555 | 691 | 815 | 1024 |
| | NRMSE | 5.89e-03 | 9.94e-03 | 1.63e-02 | 1.55e-02 | 1.20e-02 | 1.17e-02 | 1.08e-02 | 9.10e-03 | 9.48e-03 | 9.66e-03 | 8.31e-03 | 8.02e-03 | 7.00e-03 | 6.16e-03 | 6.25e-03 | 5.89e-03 | 5.85e-03 |

Table A1: Comparison of NRMSE and average sequence length $L_{\mathrm{avg,mix}}$ for Adap_Mix in ViT and SViT.

**Lid-driven cavity MHD flows**   Among the 30 cases, we kept 6 for testing. From the remaining 24 cases, we sampled 1, 3, 6, and 12 cases to assess the impact of the amount of fine-tuning data.

| Model | Metric | (16, 8, 0.05) | (32, 16, 0.05) | ps=16×16 | (32, 16, 0.0) | (16, 8, 0.02) | (16, 8, 0.1) | ps=32×32 | (16, 8, 0.3) | (16, 8, 0.0) | (32, 16, 0.2) | (16, 8, 0.01) | (32, 16, 0.5) | (32, 16, 0.1) | (32, 16, 0.3) | (32, 16, 0.8) | ps=8×8 | (16, 8, 0.2) |
|---|---|---|---|---|---|---|---|---|---|---|---|---|---|---|---|---|---|---|
| Ti-ViT-Adap_Mul | $L_\text{lin}$ | 655 | 227 | 256 | 320 | 836 | 485 | 64 | 347 | 1280 | 132 | 1001 | 97 | 167 | 114 | 79 | 1024 | 382 |
| | $L_\text{quad}$ | 67132 | 4748 | 65536 | 5120 | 67859 | 66452 | 4096 | 65900 | 69632 | 4370 | 68517 | 4231 | 4508 | 4297 | 4158 | 1048576 | 66040 |
| | NRMSE | 7.87e-03 | 1.08e-02 | 1.02e-02 | 1.13e-02 | 7.03e-03 | 8.67e-03 | 1.61e-02 | 9.85e-03 | 7.88e-03 | 1.28e-02 | 6.95e-03 | 1.36e-02 | 1.20e-02 | 1.21e-02 | 1.39e-02 | 7.06e-03 | 9.33e-03 |
| Ti-SViT-Adap_Mul | $L_\text{lin}$ | 655 | 227 | 256 | 320 | 836 | 485 | 64 | 347 | 1280 | 132 | 1001 | 97 | 167 | 114 | 79 | 1024 | 382 |
| | $L_\text{quad}$ | 67132 | 4748 | 65536 | 5120 | 67859 | 66452 | 4096 | 65900 | 69632 | 4370 | 68517 | 4231 | 4508 | 4297 | 4158 | 1048576 | 66040 |
| | NRMSE | 7.43e-03 | 9.14e-03 | 9.94e-03 | 1.03e-02 | 6.66e-03 | 8.29e-03 | 1.63e-02 | 9.25e-03 | 7.69e-03 | 1.02e-02 | 6.49e-03 | 1.24e-02 | 1.07e-02 | 1.27e-02 | 1.37e-02 | 5.89e-03 | 9.14e-03 |

Table A2: Comparison of NRMSE, $L_\text{lin}$, and $L_\text{quad}$ for Adap_Mul in ViT and SViT.

| Model | Metric | ps=8×8 | ps=16×16 | ps=32×32 | (32, 16, $g^\star = 0.1$) | (32, 16, $g^\star = 0.5$) | (32, 16, $g^\star = 0.9$) | (16, 8, $g^\star = 0.1$) | (16, 8, $g^\star = 0.5$) | (16, 8, $g^\star = 0.9$) |
|---|---|---|---|---|---|---|---|---|---|---|
| Ti-ViT-MSViT | $L_\text{avg,mix}$ | 1024 | 256 | 64 | 64 | 157 | 253 | 259 | 637 | 1018 |
| | NRMSE | 6.57e-03 | 1.04e-02 | 1.69e-02 | 1.78e-02 | 1.33e-02 | 1.05e-02 | 1.28e-02 | 9.57e-03 | 6.71e-03 |

Table A3: Comparison of NRMSE and average sequence length $L_\text{avg,mix}$ for MSViT together with constant patch size.

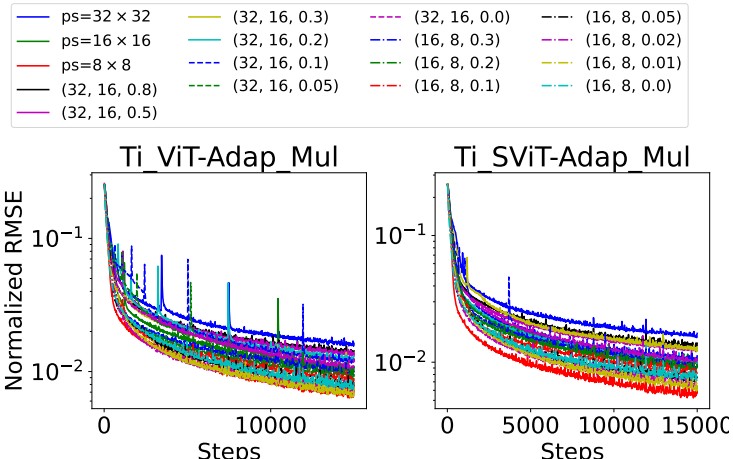

Figure A4: Comparison of training loss histories of models with adaptive tokenization Adap_Mul and constant patch sizes (ps=$32 \times 32$, ps=$16 \times 16$, and ps=$8 \times 8$) for the three spatiotemporal attention schemes (ViT and SViT) in a single colliding thermals trajectory.

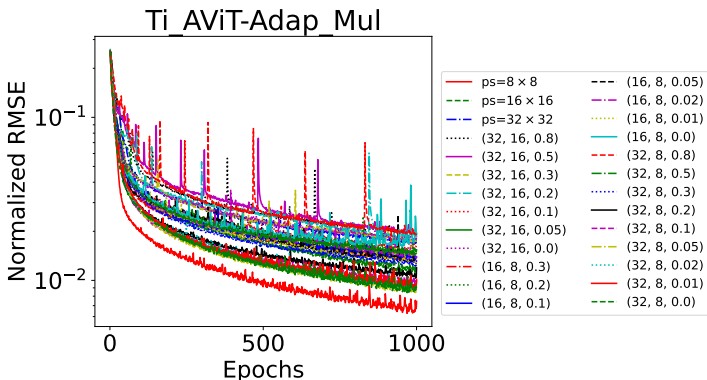

Figure A5: Comparison of training loss histories of models with adaptive tokenization Adap_Mul and constant patch sizes (ps=$32\times32$, ps=$16\times16$, and ps=$8\times8$) for AViT in a single colliding thermals trajectory.

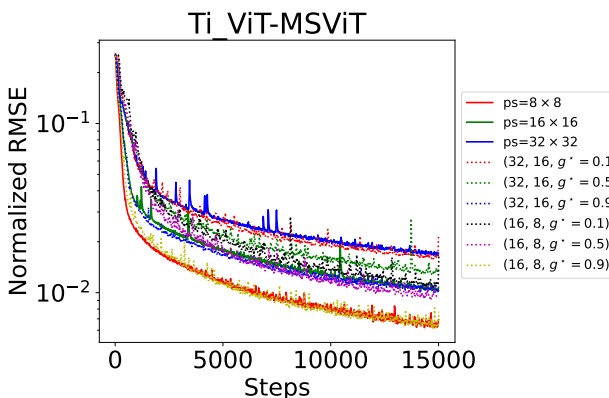

Figure A6: Comparison of training loss histories of models with adaptive tokenization MSViT (Havtorn et al., 2023) with ViT in a single colliding thermals trajectory.

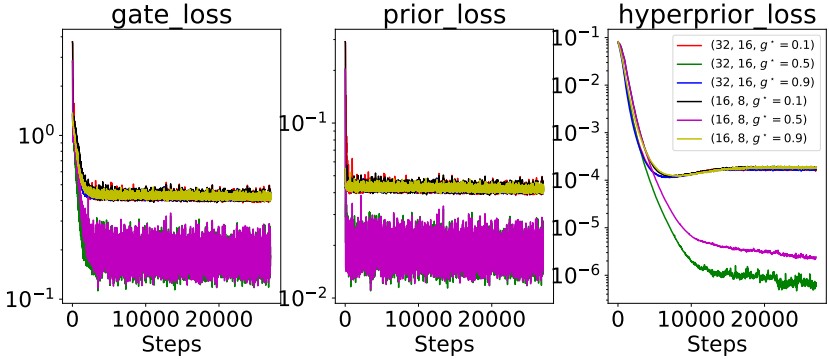

Figure A7: Training history of gate loss, prior loss, and hyperprior loss of MSViT with ViT in a single colliding thermals trajectory. The hyperparameter $\lambda$ in Equation (6) of (Havtorn et al., 2023) was set to be 10 and hence , total training loss = loss_for_prediction + gate_loss=loss_for_prediction + 10*(prior_loss + hyperprior_loss).

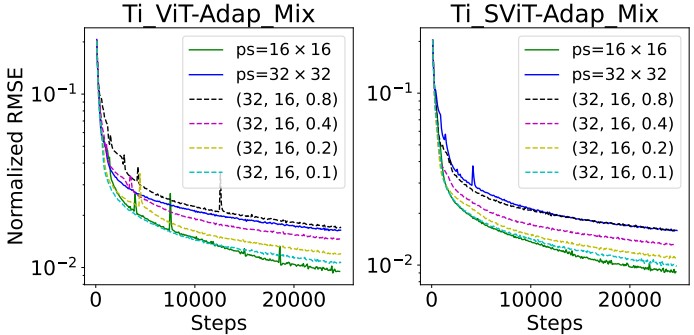

Figure A8: Comparison of training loss histories of models with adaptive tokenization Adap_Mix and constant patch sizes (ps=$32 \times 32$ and ps=$16 \times 16$) for the two spatiotemporal attention schemes (ViT and SViT) in multiple colliding thermals trajectories.

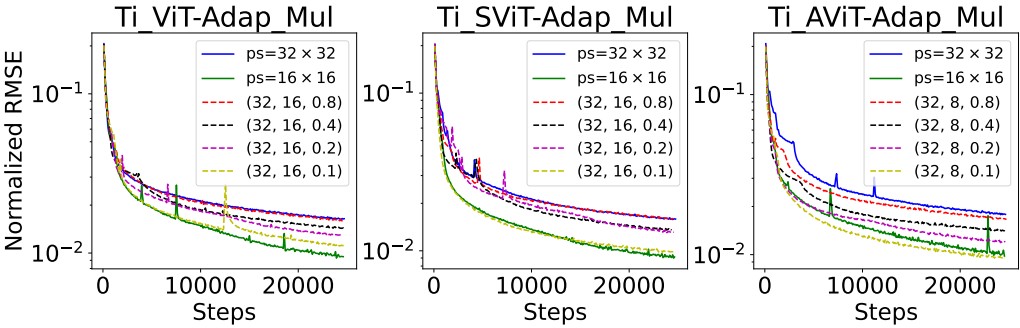

Figure A9: Comparison of training loss histories of models with adaptive tokenization Adap_Mul and constant patch sizes (ps=$32 \times 32$ and ps=$16 \times 16$) for the three spatiotemporal attention schemes (ViT, SViT, and AViT) in multiple colliding thermals trajectories.

Table A4: Cases and datasets

**Pretraining**: PDEBench Takamoto et al. (2022)

| Dataset | Variables $(C)$ | Spatiotemporal res. $(T \times H \times W)$ | $N_{\text{traj}}$ trajectories |
|---|:---:|:---:|---:|
| Shallow-water | $h$ | $101 \times 128 \times 128$ | 1,000 |
| Diffusion-reaction [diffre2d] | $\xi, \phi$ | $101 \times 128 \times 128$ | 1,000 |
| Incompressible NS | $u, v, \rho_{\text{aug}}$ | $1000 \times 512 \times 512$ | 992 |
| Compressible NS Rand-128 | $u, v, \rho, P$ | $21 \times 128 \times 128$ | 40,000 |
| Compressible NS Rand-512 | $u, v, \rho, P$ | $21 \times 512 \times 512$ | 2,000 |
| Compressible NS Turb | $u, v, \rho, P$ | $21 \times 512 \times 512$ | 2,000 |

**Fine-tuning**: colliding thermals (Section A.1.1) and lid-driven MHD (Section A.1.2)

| Dataset | Variables $(C)$ | Spatiotemporal res. $(T \times H \times W)$ | $N_{\text{traj}}$ trajectories in training |
|---|:---:|:---:|:---:|
| colliding thermals | $\rho, u, v, T$ | $1001 \times 256 \times 256$ | [1, 6, 12, 24, 48] |
| lid-driven MHD | $u, v, B_x, B_y$ | $2000 \times 128 \times 128$ | [1, 3, 6, 12, 24] |