# OpenReview forum: "MATEY: multiscale adaptive transformer models for spatiotemporal physical systems"
_TMLR — Rejected by TMLR_

### Review · Reviewer_8bEy · 2025-06-30

**Summary Of Contributions:**

The paper proposed a framework, MATEY, a multiscle transformer model leveraging adaptive mesh refinement for dynamics prediction. The author claims three key contributions in this paper. Firstly, the author compared the performance of three variants of Vision Transformers and compared their ability on challenges, such as data efficiency and computational overhead. Secondly, the author proposed an adaptive tokenization inspired by adaptive mesh refinement to handle high-resolution data. It is a promising approach to be both flexible and expressive. Thirdly, they author evaluated the physics model on out-of-distribution dataset with distinct set of variables. And find the pertaining only has limited advatange.

**Audience:**

Yes

**Broader Impact Concerns:**

There are no additional concerns on the ethical implications of the work.

**Claims And Evidence:**

No

**Requested Changes:**

I would not suggest acceptance under its current form unless the following points can be addressed.

1. The fine-tuning part has the following questions. Firstly, your model doesn't work well on the out-of-distribution dataset. By checking the contours, the best performance one still has significant visual discrepancy relative to the ground truth, and the limited fine-tuning setting actually gives very poor results visually, although it shows improvement in metrics, as shown in Figure 9 and Figure 11.

2.  The comparisons of different attention blocks seem not relevant to the main goal of this paper. I would recommend rewriting this section or moving it to supplementary parts.

3. Several figures look very similar. I would recommend adding more captions to make them more understandable.

A minor change would strengthen the work by providing more explanation on Figure 7. It is diffucult to tell the difference between the first row and the second row.

**Strengths And Weaknesses:**

The paper has the following strengths. Firstly, the adaptive tokenization is novel and mathematically rigorous on dividing patches. It is a good approach to dividing the patches adaptively based on the actual physics, which could save computational cost when the data is high resolution. Secondly, the test is pretty comprehensive and demonstrates the effectiveness of adaptive tokenization and the different fine tuning settings.

However, the paper also has the following weaknesses. First, the way of presenting needs to be improved. Several figures look quite similar and confusing without cross-checking several components in the papers. Second, the experiment to evaluate the performance of different attention modules seems unnecessary in the main part of this paper, as it didn't introduce any novelty. Third, the fine-tuning part of the result is not convincing. There is almost no improvement in the pretrained model if fine-tuning all the parameters, which makes it not worth pretraining a model. Moreover, the error in the fine-tuning is still relatively large, as checked by the contour, making it hard to justify the usefulness of such a setting.

---

> ### Author Response · Authors · 2025-07-17
> **response to 8bEy**
>
> 1. The fine-tuning part has the following questions. Firstly, your model doesn't work well on the out-of-distribution dataset. By checking the contours, the best performance one still has significant visual discrepancy relative to the ground truth, and the limited fine-tuning setting actually gives very poor results visually, although it shows improvement in metrics, as shown in Figure 9 and Figure 11.
>
> Our response:
> Yes, we agree that our pretrained model has limited advantage on out-of-distribution datasets. We think this is a notable point that is not well-reflected in the literature.
>
> There are a few studies that focus on fine-tuning tasks on same physical systems that were used for pretraining and reported promising results. But a more broad and often model usage scenario, where one would expect a foundation model-style approach to be most important, would be target systems that have new distinct physics.
>
> While challenging and with preliminary results, our fine-tuning tasks are designed to provide a valuable first set of data points to the SciML community in this more realistic usage scenario.
>
> Given this purpose, we did not try to optimize for the performance of each individual case.
>
> 2. The comparisons of different attention blocks seem not relevant to the main goal of this paper. I would recommend rewriting this section or moving it to supplementary parts.
>
> Our response:
> Thank you for your comment. We agree that the connection between attention comparison and the main goal of the paper could be clarified further.
>
> Our intention with this section was to investigate the performance of different attention blocks, designed for handling spatiotemporal sequences in video data, in multiscale physical data. It serves as a baseline of our proposed approach:  we started from these then added on adaptive tokenization. As our results (particularly with AViT) indicate, adaptive tokenization performance varies significantly between spatiotemporal attention schemes, necessitating a clear evaluation of the baseline performance of the spatiotemporal attention schemes themselves to put the results in appropriate context.  This two-part evaluation is necessary to prepare for the ultimate consideration of long 3D sequences in future work.
>
>
> 3. Several figures look very similar. I would recommend adding more captions to make them more understandable.
>
> Our response:
> We’ve updated figure captions in the revised manuscript.
>
> 4. A minor change would strengthen the work by providing more explanation on Figure 7. It is diffucult to tell the difference between the first row and the second row.
>
> Our response:
> Thanks for pointing this out. As suggested, we’ve added two row-captions and more explanations in caption to improve the readability.

---

### Review · Reviewer_uaXt · 2025-06-30

**Summary Of Contributions:**

Vision Transformers (ViTs) operate by dividing the images into usually non-overlapping small regions called patches (aka tokens). Therefore, for a given image resolution, we would get a fixed number of tokens. There are works in the literature that employ variable patch sizes to increase model expressivity.

This work proposes two adaptive tokenization methods as core contributions, namely Adaptive Multi-resolution and Adaptive Mixed Resolution, where the patch sizes are adjusted (coarse (32$\times$32) --> fine (16$\times$16)) based on the informativeness of the underlying local features, which is captured via the patch variance. The authors show that this allows the model to allocate computational resources more efficiently, focusing on information-rich regions while maintaining a coarse representation elsewhere, eventually reconstructing a coherent, multi-resolution output.

The work also explores the benefits of pretraining on diverse datasets from PDEBench for out-of-distribution generalization to totally unseen phenomena of thermal and electromagnetic components.

**Audience:**

Yes

**Broader Impact Concerns:**

The proposed research improves the state-of-the-art capabilities of vision transformers, and the reviewer sees this as falling under advancing a field of research. However, the applicability to physical systems might find an adverse use case. Hence, the authors are requested to add a section on this in the revision.

**Claims And Evidence:**

Yes

**Requested Changes:**

**Comparison with Baselines**: The authors identify MSViT as the most related work, but do not include it in the experimental comparisons. A direct comparison is crucial for contextualizing the performance of MATEY and substantiating its advantages over the state-of-the-art in multi-scale transformers.

**Applicability to 3D Spatial Problems**: The presented framework is limited to 2D spatial domains. Given that many critical scientific problems (e.g., in fluid dynamics, climate science) are inherently 3D, the paper would be significantly strengthened by a discussion on the applicability and expected scalability of the proposed adaptive mechanism to 3D spatial data. Can the authors comment on the potential challenges (e.g., computational cost of 3D patch variance, memory constraints) and necessary architectural modifications?

**Clarity on Hyperparameter Tuning**: The variance threshold, $\gamma_{\text{sts}}$, is a critical hyperparameter that governs the model's adaptive behavior. The paper does not specify how this value was selected (e.g., $\gamma_{\text{sts}} = 0.2$ in Section 4.2). Or was it chosen only for representational purposes? In general, can the authors comment about what values are optimal to be chosen for a specific problem, since not everyone can afford a sweep across a wide range of values as you have done in Figures 5-7? A discussion on the tuning process and the sensitivity of the model's performance to this parameter would enhance the paper's usefulness.


**Clarity and Notational Consistency**

**Figure 1 (SViT and ViT diagrams**: There is a typo in the second row of the ViT token representation. Following the established $(t, x, y)$ coordinate system, the last element should be (1,npx,2), not (1,npy,2) as it is currently. This should be corrected.

**Definition of Spatial Dimensions**: The text states, "Specifically, letting H and W denote the resolution in the x and y directions, respectively,". This should be corrected to "Specifically, letting W and H denote the resolution in the x and y directions, respectively," because as can be seen in Fig. 1, the indices vary along the x axis, denoting the width and likewise for height.

**Tensor Dimensions**: To be consistent with the corrected Figure 1 and the text, the definition of the initial token tensor $Z^0 \in R^{nt \times npx \times npy \times C_{emb}}$ should be fixed, and accordingly the definitions of $npx$ and $npy$.

**Equation before (5)**: In the definition of $Z^{L_{sts},i}$, the final term $p_{x_1}/p_{x_{sts}} \times p_{y_1}/p_{y_{sts}}$ is missing the `sts` subscript in the variable $Z^L$.

**Section 3 Title**: The authors should perhaps consider a more meaningful title for section 3.

**Patch Indices $(i, j)$**: For clarity, please explicitly state that $(i, j)$ in Equations (3) and (4) represent the 2D spatial indices on the patch grid. For example, adding a sentence like, "Here, $(i, j)$ identifies the patch's coordinates on the $npx_1 \times npy_1$ grid."

**Conclusion Section**: The paper currently lacks a dedicated "Conclusion" section to summarize the key findings, contributions, and future directions.

-------------------

**Typos and Grammar**: Please correct the following:

 - "sciML" $\rightarrow$ "SciML"
 - "random initialized models" $\rightarrow$ "randomly initialized models"
 - "The govern equations" $\rightarrow$ "The governing equations"

**Abbreviation**: The term "multiple physics pretraining" should be explicitly abbreviated as (MPP) on its first use and then referred to as MPP thereafter.

**CNN vs. Convolutional Block**: The text states, "the tokenization module is implemented as a convolutional neural network (CNN)". As a CNN typically implies a network of at least two layers, and given the context of a decoding "postprocessor," it seems more accurate to state that it is "implemented as a convolutional block."

**Redundancy in Plots**: In Figures 3 and 10, the y-axis labels are "NRMSE loss." Since NRMSE is an error metric itself, the word "loss" is redundant and should be removed (e.g., label as "NRMSE"). Similarly, "NRMSE errors" in the text should be shortened to "NRMSE."

----------------

**Questions for the Authors**

  - Missing AViT for Adap--Mix: Why is there no AViT for the Adap_Mix strategy (cf. Fig. 2)?
  - Why was the same investigation not extended to 3D spatial data? Can the authors comment on the applicability and scalability of the current approach to 3D?
  - Why is the model called "MATEY", and where does it come from? Can the authors add a note on that?

  - Will the source code and pretrained models be made publicly available? Releasing the code would ensure the reproducibility of the results and facilitate future research.

**Strengths And Weaknesses:**

Strengths:

- The paper is well-written and fairly easy to understand, although the notations and naming conventions could be improved.

- Content-aware, adaptive tokenization scheme advances ViTs for spatiotemporal data, directly addressing the critical challenge of computational scalability in modeling high-resolution physical systems.

- The tokenization strategies determined using patch variance are simple to implement and seem to be effective.

- The steps are provided in a detailed manner to reconstruct the multi-resolution outputs.

- Conducts extensive experiments across two different ViT variants -- SViT and AViT, across three model scales (Tiny, Small, Base), and includes a sweep across different values for the patch refinement hyperparameter ($\gamma_{sts}$).

- The paper demonstrates the effectiveness of the pretrained representations for fine-tuning on challenging problems, including MHD flows and electromagnetics. The investigation into out-of-distribution generalization through multiple physics pretraining provides valuable insights for the broader SciML.

Weaknesses:

- The paper assumes that phenomena requiring adaptive tokenization strategies are present in the physical problems.

- The code is not open-source, and it is unclear whether it will be made public or not upon acceptance.

---

> ### Author Response · Authors · 2025-07-17
> **response to uaXt -part 1**
>
> 1. The paper assumes that phenomena requiring adaptive tokenization strategies are present in the physical problems.
> The code is not open-source, and it is unclear whether it will be made public or not upon acceptance.
>
> Our response:
> We’d like to clarify that our adaptive tokenization methods are most suitable and, indeed, are designed for physical systems demonstrating spatial inhomogeneity. Such physical systems cover a very wide range of application scenarios such as ocean flows, multiphase flows, and reacting flows – making our method broadly applicable in the SciML field. These spatial heterogeneities allow us to prioritize important areas of the physical domain, given a computing budget, while preserving accuracy. Text emphasizing this clarification has been added to the revised paper.
>
> We will make the code open source upon paper publication.
>
> 2. Comparison with Baselines: The authors identify MSViT as the most related work, but do not include it in the experimental comparisons.
>
> Our response:
> We agree that adding a MSViT baseline comparison would make our paper stronger, and we’ve added a new subsection with Figure 8 in section 4.2 with a direct comparison in the revised manuscript.
>
> The MSViT codebase has not been released.  Thus, we’ve implemented the gating neural network-controlled tokenization scheme in MSViT based on the description in (Havtorn et al., 2023)  and conducted a set of comparison runs with our adaptive tokenization schemes. We’ll release our implementation of MSViT along with the two proposed adaptive tokenization schemes.
>
> The results show that Adap_Mix is more efficient than MSViT, achieving higher accuracy with shorter sequences. Adap_Mix refines patches directly based on input data, while MSViT uses a jointly trained gating NN that initially refines all patches (hence high $L_\textup{avg,mix}$ values). Thus, MSViT runs with longer sequences during training until the gate converges, which happens after the hyperprior distribution stabilizes.
>
> 3. Applicability to 3D Spatial Problems: The presented framework is limited to 2D spatial domains. Given that many critical scientific problems (e.g., in fluid dynamics, climate science) are inherently 3D, the paper would be significantly strengthened by a discussion on the applicability and expected scalability to 3D data. Can the authors comment on the potential challenges and necessary architectural modifications?
>
> Our response:
> The algorithms themselves have no limitations in extending to 3D, and the computational cost of patch variance would not be an issue. Rather, the potential constraints mainly arise from the generic memory and attention computation needs of the long token sequence lengths associated with 3D systems. Fully addressing the 3D setting would require a separate work. We are currently conducting work dealing with extremely long sequence length in high-resolution 3D data with a hierarchical transformer and a sequence parallelism algorithm. Future work includes combine the two techniques together. We’ve added the discussion of the potential computational challenges when extending to 3D in the future work section of the revised manuscript.
>
> While the paper focuses on spatially 2D experimental demonstrations, the adaptive algorithms are applicable to 3D real-worlds problems with limitations. In practice, as attention cost scales quadratically with the sequence lengths, an extension to 3D data would be still much more resource-intensive. This has motivated our work to identify additional methods to address these challenges, but they fall well outside of the scope of the current study.

---

> > ### Author Response · Authors · 2025-07-17
> > **response to uaXt -part 2**
> >
> > 4. Clarity on Hyperparameter Tuning: The variance threshold... Or was it chosen only for representational purposes? In general, can the authors comment about what values are optimal to be chosen for a specific problem, ...? A discussion on the tuning process and the sensitivity of the model's performance to this parameter would enhance the paper's usefulness.
> >
> > Our response:
> > Yes, the gamma value (0.2) in Figure 4 was chosen solely for illustrating the idea. In practice, we keep it as a user-specified parameter so that users can balance accuracy and cost when designing experiments.
> >
> > Generally, small gamma values produce better accuracy but at a somewhat higher computational cost (depending on the choice of tokenization schemes, user settings, and the dataset). For Adap_Mul, the attention cost increase for a smaller gamma is negligible, but the accuracy gain is substantial; accordingly, we would recommend choosing a lower gamma value.
> >
> > For Adap_Mix, the sequence length (and hence the cost) will grow more quickly with smaller gamma, while still providing significantly speedup. We would suggest the user to decide the gamma value based on the available computing resources: that is, that they identify what is the sequence length they could afford from based on memory and compute allocation limitations, and then work backward to select gamma based on the dataset and the set of patch sizes that they intend to use.
> >
> > We’ve added a discussion of this question to the revised manuscript in the end of page 10.
> >
> >
> > 5. Clarity and Notational Consistency
> >
> > 5.1 Figure 1 (SViT and ViT diagrams: There is a typo ...)
> >
> > Our response:
> > Thank you for catching typo and notation inconsistency. We’ve updated Figure 1 to have H (height) and W (weight) represent the resolution in x and y directions, respectively.
> >
> > 5.2 Tensor Dimensions
> >
> > Our response:
> > Thank you for catching this! We have fixed this in the revised version.
> >
> > 5.3 Section 3 Title: The authors should perhaps consider a more meaningful title for section 3.
> >
> > Our response:
> > We’ve updated the title to “Methods” in the revised version and better organized the content into two subsections.
> >
> > 5.4 Patch Indices : For clarity, please explicitly state that  in Equations (3) and (4) represent the 2D spatial indices on the patch grid.
> >
> > Our response:
> > We appreciate this suggestion, agree it make a lot of sense, and have revised the draft accordingly.
> >
> > 5.5 Conclusion Section: The paper currently lacks a dedicated "Conclusion" section to summarize the key findings, contributions, and future directions.
> >
> > Our response:
> > We’ve updated the summarization of key findings and contributions in the previous “Discussion” section to be a “Conclusion” section with a paragraph on future directions included.
> >
> > 6. Typos and Grammar
> >
> > Our response:
> > We appreciate the reviewer’s careful read of the manuscript and have fixed these mistakes in the revised version.
> >
> >
> > 7. Missing AViT for Adap--Mix: Why is there no AViT for the Adap_Mix strategy (cf. Fig. 2)?
> >
> > Our response:
> > Adap_Mix does not support AViT. Because AViT requires the patches to be nicely distributed as a structure grid in each direction, while the grid structure is not preserved in Adap_Mix when patches at different resolutions are mixed.
> >
> > 8. Why was the same investigation not extended to 3D spatial data? Can the authors comment on the applicability and scalability of the current approach to 3D?
> >
> > Our response:
> > The algorithms are applicable to 3D. We focus on 2D simply because of the computational costs for conducting parameter sweep quickly become quite expensive. We’d like to test the methods first in 2D examples that are cost effective and demonstrate the effectiveness of the approach before extending it to 3D. Conceptually, we do not foresee any additional algorithmic/implementation challenges in 3D from the adaptive tokenization side. On the other hand, scalability issues will inevitably arise from the general long-sequence challenge that all transformer models have. We’re currently working on this with novel hierarchical transformer architecture and sequence parallelism. We have future work planned to combine the two techniques together for 3D.
> >
> > 9. Why is the model called "MATEY", and where does it come from? Can the authors add a note on that?
> >
> > Our response:
> > MATEY (the full name: ARRR, MATEY) is a Multiscale AdapTivE trustworthY transformer-based codebase and model for spatiotemporal physical systems, which is currently under active development. ARRR means accuracy and reliability, robustness, and resilience, which represents our long-term model performance target.
> >
> > 10. Will the source code and pretrained models be made publicly available? Releasing the code would ensure the reproducibility of the results and facilitate future research.
> >
> > Our response:
> > Yes, we’ll make these openly available upon paper acceptance.
> >
> > 11. Broader Impact Concerns
> >
> > Our response:
> > We have added a broader impact statement to the revised manuscript.

---

### Review · Reviewer_o5Eb · 2025-07-05

**Summary Of Contributions:**

The submission introduces MATEY, a vision transformer (ViT)-based model designed for multiscale spatiotemporal physical systems. It proposes two adaptive tokenization schemes (Adapt_Mul and Adapt_Mix) that dynamically adjust patch sizes based on local features, aiming to reduce computational costs compared to uniform patch refinement. Additionally, it explores spatiotemporal attention schemes (ViT, SViT, AViT) with varying degrees of decoupling and evaluates their efficiency. The authors claim improved accuracy with adaptive tokenization and demonstrate fine-tuning performance on out-of-distribution datasets (colliding thermals and magnetohydrodynamics) using pretrained models on PDEBench data. However, the contributions lack novelty, and the experimental validation is insufficient to support the claims, with several methodological and clarity issues undermining the work’s impact.

**Audience:**

Yes

**Broader Impact Concerns:**

The submission lacks a Broader Impact Statement, which is a significant oversight given the potential applications in energy generation, earth sciences, and propulsion systems. The paper does not discuss the risks of deploying MATEY in safety-critical systems, where inaccurate predictions could have severe consequences (e.g., in weather forecasting or power grid management). A Broader Impact Statement addressing these issues is necessary to ensure responsible deployment.

**Claims And Evidence:**

No

**Requested Changes:**

1. Provide a detailed comparison with existing tokenization schemes (e.g., MSViT, A-ViT) and attention mechanisms (e.g., Swin Transformer, MViT2).

2. Reconcile the conflicting statements about token sequence length in the abstract and main text. Provide quantitative evidence (e.g., sequence length metrics across experiments) to support efficiency claims.

3. Include comparisons with state-of-the-art baselines and evaluate adaptive tokenization on multiple trajectories and datasets. Report results for larger model sizes and higher-resolution inputs to demonstrate scalability.

4. Provide a comprehensive description of hyperparameters, training protocols, and computational resources. Include a publicly accessible codebase or pseudocode for adaptive tokenization schemes.

5. Conduct ablation studies to isolate the contributions of adaptive tokenization versus attention schemes.

**Strengths And Weaknesses:**

**Strengths**

1. The idea of dynamically adjusting patch sizes based on local features is conceptually interesting and aligns with adaptive mesh refinement techniques.
2. Testing fine-tuning on datasets with different physics (colliding thermals and MHD) is a valuable attempt to assess model generalizability.

**Weaknesses**

1. The adaptive tokenization schemes are incremental extensions of existing methods (e.g., MSViT by Havtorn et al., 2023). The authors fail to clearly differentiate MATEY’s tokenization from prior work, and the attention schemes (ViT, SViT, AViT) are largely based on established frameworks like axial attention (Ho et al., 2019).

2. The abstract claims adaptive tokenization achieves “improved accuracy without significantly increasing the length of the token sequence,” while the introduction and experiments (e.g., Section 4.2) suggest increased sequence lengths with Adapt_Mix, which undermine the efficiency claims.

3. The experiments are limited in scope, with only one trajectory evaluated for adaptive tokenization (Section 4.2) and small model sizes (Tiny, Small, Base).

4. Key hyperparameters (e.g., $\gamma_{\text{sts}}$, patch sizes) and implementation details (e.g., training protocols, computational resources) are inadequately described.

---

> ### Author Response · Authors · 2025-07-17
> **response to o5Eb - part1**
>
> 1. The adaptive tokenization schemes are incremental extensions of existing methods (e.g., MSViT by Havtorn et al., 2023). The authors fail to clearly differentiate MATEY’s tokenization from prior work, and the attention schemes (ViT, SViT, AViT) are largely based on established frameworks like axial attention (Ho et al., 2019).
>
> Our response:
> We would like to clarify that the novelty of this work lies in the two new adaptive tokenization schemes, Adap_Mul and Adap_Mix, which are introduced in Section 3.2 in the revised manuscript. These two adaptive tokenization schemes are different from the existing adaptive tokenization scheme MSViT : (1) while Adap_Mix and MSViT are both  mixed-scale adaptive tokenization methods,  Adap_Mix does not rely on an additional neural network and potentially unknow prior distributions to control the refinement; and (2) Adap_Mul  is fundamentally more computing efficient and different from MSViT, as it treats tokens at different scales separately.
>
> The effectiveness of these two tokenization schemes in three spatiotemporal attention schemes was illustrated in Figures 5, 6, and 7. Adap_Mix provides better predictive accuracy, likely due to considering cross-scale correlations, and guarantees convergence toward the solution with uniformly refined tokens. In contrast, Adap_Mul is dramatically more cost effective for attention operations with quadratic complexity and easier to implement than Adap_Mix. AViT does not interact well with adaptive tokenization approaches when the STS sequence length is too short.
>
> We also added a comparison between Adap_Mix with MSViT in Figure 8. The results show that Adap_Mix is more efficient than MSViT, achieving better accuracy with shorter sequence lengths. This efficiency is due to Adap_Mix directly selecting patch refinement based on input data, whereas MSViT relies on a gating NN jointly trained with the model. The gate begins with an initial condition that refines all patches (i.e., high initial Lavg,mix values). During training, MSViT operates with longer sequence length until the gating NN converges, which occurs after the convergence of the hyperprior distribution.
>
> As for the attention mechanisms, we agree that ViT, SViT, and AViT are commonly used mechanisms from existing work, as discussed in Section 2 (Related work). We include them in this work to demonstrate the performance of the adaptive tokenization schemes on these common attention mechanisms. In addition, to our knowledge, the performance and data efficiency result reported in Section 4.1 is the first comparison between these attention mechanisms on physical system applications, which is a standalone contribution to the SciML community.
>
> We have revised the manuscript in several places to more clearly express the novelty of the adaptive tokenization methods introduced and to clarify why the spatiotemporal attention schemes are presented and evaluated.
>
> 2. The abstract claims adaptive tokenization achieves “improved accuracy without significantly increasing the length of the token sequence,” while the introduction and experiments (e.g., Section 4.2) suggest increased sequence lengths with Adapt_Mix, which undermine the efficiency claims.
>
> Our response:
> Thanks for pointing this out. We acknowledge that original statement is not sufficiently clear. Our adaptive tokenization methods allow for dynamically adjusting patch resolution based on local features. Figure 5 in Section 4.2 shows that the token sequence lengths from Adap_Mix are between the coarse patch and the fully refined cases, dependent on the thresholding parameter. The results there further illustrate that Adap_Mix achieves comparable accuracy to the fully refined case with much shorter token sequences in average. The comparison between Adap_Mix and MSViT presented in Figure 8 (see below for detailed discussion) also confirms the computational advantage of Adap_Mix. We have revised the abstract to clarify this point.
>
> 3. The experiments are limited in scope, with only one trajectory evaluated for adaptive tokenization (Section 4.2) and small model sizes (Tiny, Small, Base).
>
> Our response:
> We’ve added Figures 9 and 10, which report results from a set of experiments on multiple trajectories (512 for training and 64 for testing), into Section 4.2 in the revised manuscript. These results resemble the ones tested on a single trajectory in Figures 5, 6, and 7.
>
> 4. Key hyperparameters (e.g., , patch sizes) and implementation details (e.g., training protocols, computational resources) are inadequately described.
>
> Our response:
> To keep the story concise, the implementation details are given in supplementary materials (see section A.2). We will open source our code implementation and settings for the test runs upon paper acceptance.

---

> > ### Author Response · Authors · 2025-07-17
> > **response to o5Eb - part2**
> >
> > 5. Provide a detailed comparison with existing tokenization schemes (e.g., MSViT, A-ViT) and attention mechanisms (e.g., Swin Transformer, MViT2).
> >
> > Our response:
> > We acknowledge that a detailed comparison would be helpful. As suggested, we have compared Adap_Mix to MSViT and included the comparison results in Figure 8, Section 4.2 in the revised manuscript.  The results indicate that, with comparable token sequence length, Adap_Mix leads to lower prediction error than MSViT. Because the codebase for MSViT has not been released, we implemented the MSViT adaptive tokenization following the description in (Havtorn et al., 2023) with the batch-shaping loss implementation provided therein. We plan to release our MSViT implementations and settings together in our codebase.
> >
> > We did not compare the adaptive tokenization schemes to A-ViT, because A-ViT is limited to pruning after training and does not reduce the training time. Consequently, A-ViT is not applicable for comparison in this study.
> >
> > While we agree that comparison with other attention mechanisms like Swin Transformer and MViTv2 is interesting, we consider it in the scope of future work given that the proposed adaptive tokenization schemes have been evaluated with three different attention mechanisms (ViT, SViT, AViT).
> >
> > 6. Reconcile the conflicting statements about token sequence length in the abstract and main text. Provide quantitative evidence (e.g., sequence length metrics across experiments) to support efficiency claims.
> >
> > Our response:
> > Figs. 5-7 show predictive error at varying sequence lengths (or cost associated with sequence lengths). We’ve also added Tables A1, A2, and A3 to report the quantitative results in the supplementary materials.
> >
> > 7. Include comparisons with state-of-the-art baselines and evaluate adaptive tokenization on multiple trajectories and datasets. Report results for larger model sizes and higher-resolution inputs to demonstrate scalability.
> >
> > Our response:
> > Thank you for these suggestions. In response, we’ve added both MSViT baseline results (Figure 8) and adaptive tokenization on multiple trajectories (Figures 9 and 10) in the revised manuscript. Those additions make our findings stronger and more general.
> >
> > As for scalability demonstrations on more datasets, larger model sizes, and higher-resolution samples would be beyond the scope and deserve a separate study, given the computing resources required. Our focus for this study was to demonstrate a proof-of-concept for our approach that justified future extension to much larger models and larger, more complex datasets. Building on this work, we’re currently conducting a study that leverages a novel hierarchical transformer architecture and sequence parallelism for 3D high-resolution fluid dynamics data. Future work is planned to combine the techniques in two papers together for real-world 3D systems.
> >
> > 8. Provide a comprehensive description of hyperparameters, training protocols, and computational resources. Include a publicly accessible codebase or pseudocode for adaptive tokenization schemes.
> >
> > Our response:
> > Descriptions of hyperparameters, training procedures, and computational resources used are presented in Section A.2 of supplementary materials. We will release our codebase with adaptive tokenization implementations upon paper publication.
> >
> > 9. Conduct ablation studies to isolate the contributions of adaptive tokenization versus attention schemes.
> >
> > Our response:
> > We acknowledge the importance of clearly isolating the relative contributions to computational efficiency and model utility of the adaptive tokenization and attention schemes. Because there are only two techniques considered, we determined that simply showing the adaptive tokenization results separately for each attention scheme was clearer than presenting the same results in the context of an ablation study. Consequently, section 4.1 addresses the baseline spatiotemporal attention scheme performance and section 4.2 considers the addition of adaptive tokenization to each of the spatiotemporal attention schemes.
> >
> > 10.  Broader Impact Concerns: The submission lacks a Broader Impact Statement, which is a significant oversight given the potential applications in energy generation, earth sciences, and propulsion systems. The paper does not discuss the risks of deploying MATEY in safety-critical systems, where inaccurate predictions could have severe consequences (e.g., in weather forecasting or power grid management). A Broader Impact Statement addressing these issues is necessary to ensure responsible deployment.
> >
> > Our response:
> > We acknowledge the omission of a Broader Impact Statement in the submission and have added it in the revised version.

---

### Author Response · Authors · 2025-07-17
**Response to reviewer' comments**

We thank the reviewers for their constructive review comments, which have helped improve our revised manuscripts. We have updated the manuscript to address reviewers’ points, with all changes highlighted in blue. We will provide our detailed response now.

---

### Decision · Action_Editor_jTZJ · 2025-09-18

**Recommendation:** Reject

**Additional Comments:**

As noted by reviewer 8bEy, the core focus of the paper is not clear and the manuscript could be revised with a clear focus on a the key problematic. The method is mostly tested (and developed) to handle multi-scale phenomena, if that is the case, the organization, intro, abstract and experiments of the paper should clearly be centered around that.

Comparison with other methods: the field of AI for science is very active and many approaches have been proposed, with numerous works evaluating on PDEBench in particular. As such I would expect a comparison with the existing SOTA (transformer based foundation models, neural operators) and a thorough review of related works.

Given the incremental nature of the method (The adaptive tokenization schemes are incremental extensions of existing methods, as noted by reviewer o5Eb), we expect a much more thorough discussion of previous works and related literature as well a substantial empirical evaluation that should include comparisons to the state-of-the-art. The experimental setting in its current form is too limited.
The author could also anonymously submit code to reproduce the experiments along with the manuscript, or at least revise the description of the method to enable reproducibility.

Overall, there is a clear consensus that the paper in its current form is not ready for publication. The changes recommended would make the work much more impactful.

**Audience:**

Yes

**Audience Explanation:**

While the paper in its current state somewhat lacks focus (and review of relevant works), the topic is of interest for the AI4science community.

**Claims And Evidence:**

No

**Claims Explanation:**

Comparison with other methods: the field of AI for science is very active and many approaches have been proposed, with numerous works evaluating on PDEBench in particular. As such I would expect a comparison with the existing SOTA (transformer based foundation models, neural operators) and a thorough review of related works. It is hard to fairly assess the methods without these discussions and comparison.

Given the incremental nature of the method (The adaptive tokenization schemes are incremental extensions of existing methods, as noted by reviewer o5Eb), we expect a much more thorough discussion of previous works and related literature as well a substantial empirical evaluation that should include comparisons to the state-of-the-art. The experimental setting in its current form is too limited.

The author could also anonymously submit code to reproduce the experiments along with the manuscript, or at least revise the description of the method to enable reproducibility.

**Resubmission Of Major Revision:**

The authors may consider submitting a major revision at a later time.